



# Comparison of the GRUAN data products for Meisei RS-11G and Vaisala RS92-SGP radiosondes at Tateno (36.06°N, 140.13°E), Japan

Eriko Kobayashi[1], Shunsuke Hoshino[1], Masami Iwabuchi[2], Takuji Sugidachi[3], Kensaku Shimizu[3] and Masatomo Fujiwara[4]

[1]Aerological Observatory, 1-2 Nagamine, Tsukuba-shi, Ibaraki, 305-0052, Japan
[2]Japan Meteorological Agency, 1-3-4 Otemachi, Chiyoda-ku, Tokyo, 100-8122, Japan
[3]Meisei Electric Co., Ltd., 2223 Naganumamachi, Isesaki-shi, Gunma, 372-8585, Japan
[4]Faculty of Environmental Earth Science, Hokkaido University, Kita 10 Nishi 5, Kita-ku, Sapporo, 060-0810, Japan

*Correspondence to*: Eriko Kobayashi (eriko-kobayashi@met.kishou.go.jp)

**Abstract.** A total of 87 dual flights of Meisei RS-11G radiosondes and Vaisala RS92-SGP radiosondes were carried out at the Aerological Observatory of the Japan Meteorological Agency (36.06ºN, 140.13ºE, 25.2 m) from April 2015 to June 2017. Global Climate Observing System (GCOS) Reference Upper-Air Network (GRUAN) data products from both sets of radiosonde data for 52 flights were subsequently created using a documented processing program along with the provision of optimal estimates for measurement uncertainty. The authors then quantified differences in the performance of the radiosondes using GRUAN data products. The temperature measurements of RS-11G were 0.4 K lower than those of RS92-SGP in the stratosphere during daytime observation. The relative humidity measurements of RS-11G were 2%RH lower than those of RS92-SGP under 90–100%RH conditions, while RS-11G gave 5%RH higher values than RS92-SGP under ≤ 50%RH conditions. The results from a dual flight of RS-11G and a cryogenic frostpoint hygrometer (CFH) also showed that RS-11G gave 1–10%RH higher values than the CFH in the troposphere. The authors additionally investigated the RS-11G minus RS92-SGP difference of temperature and relative humidity based on combined uncertainties to clarify major influences behind the difference. It was found that temperature differences in the stratosphere during daytime observation were within the range of uncertainty (k = 2), and that sensor orientation is the major source of uncertainty in RS92-SGP temperature measurement, while sensor albedo is the major source for RS-11G. The relative humidity difference in the troposphere was larger than the uncertainty (k = 2) after the radiosondes had passed through the cloud layer, and temperature-humidity dependence correction was the major source of uncertainty in RS-11G relative humidity measurement. Uncertainties for all soundings were also statistically investigated. Most night-time temperature differences for pressures of > 10 hPa were in agreement, while relative humidity differences in the middle troposphere exhibited significant differences. Around half of all daytime temperature differences at pressures of ≤ 150 hPa and relative humidity differences around the 500 hPa level were not in agreement.



# 1 Introduction

The Aerological Observatory of the Japan Meteorological Agency (JMA) (location: Tateno, 36.06ºN, 140.13ºE, 25.2 m above mean sea level) has played a leading role in the operation of all JMA's radiosonde stations since its establishment in 1920. The Tateno station was chosen as a candidate site for the Global Climate Observation System (GCOS) Reference Upper Air Network (GRUAN; Seidel et al., 2009; Bodeker et al., 2016) in 2009, and was certified as a GRUAN site in 2018. The Vaisala RS92-SGP radiosonde (referred to here as RS92; Dirksen et al., 2014) was used for routine observation at the site from December 2009 to June 2013 (Kobayashi et al., 2015), after which it was replaced with the Meisei RS-11G radiosonde (Kizu et al., 2018). The Meisei iMS-100 radiosonde (Kizu et al., 2018) was also introduced in September 2017. RS-11G is equipped with a thermistor, a capacitive relative humidity (RH) sensor, a Global Positioning System (GPS) receiver for monitoring height, pressure and horizontal wind, and a transmitter at 400 MHz (Kizu et al., 2018). Compared with the previous-generation Meisei RS-06G radiosonde (Nash et al., 2011), temperature and RH measurement has been improved via hardware and software upgrades. RS-11G is used at a variety of JMA stations, at stations of other meteorological service providers, and by numerous research institutes and universities.

GRUAN will provide long-term, high-quality climate data records for levels ranging from the surface to the troposphere to the stratosphere (Seidel et al., 2009; Bodeker et al., 2016). GRUAN data products (GDPs) will be open, documented in peer-reviewed literature and traceable to SI standards, and will have the best possible estimates of vertically resolved measurement uncertainties (Dirksen et al., 2014). When measurement systems, including instrument types are changed, any systematic biases between the old and new systems need to be characterized (GCOS-134, 2009; GCOS-170, 2013). In this context, JMA previously made several changes in radiosonde types for improved upper-air measurement with state-of-the-art technology (Kizu et al., 2018), and habitually makes dual flights of old and new radiosondes to characterize these measurements (JMA Aerological Division, 1983; Sakoda et al., 1999; Kobayashi et al., 2012). Kobayashi et al. (2012) reported results from 103 dual flights of the Meisei RS2-91 rawinsonde and RS92 when the latter was introduced at Tateno.

Following the change in routine radiosonde type from RS92 to RS-11G at Tateno in July 2013, a total of 87 weekly dual flights of RS-11G and RS92 were conducted from April 2015 to June 2017 (avoiding the July to mid-September period when there is an elevated chance that a radiosonde may fall into the densely populated Tokyo metropolitan area). The GRUAN data product made from RS92 measurements at Tateno was created at the GRUAN Lead Centre, and is available on the GRUAN website at https://www.gruan.org/. The GRUAN data product made from RS-11G measurements was created at Tateno and submitted to the GRUAN Lead Centre, and will be available on the GRUAN website when this paper is published. A novel aspect of GRUAN data products is that vertically resolved uncertainty estimates and metadata are provided for each sounding. Quantitative comparisons of GRUAN data products based on data from these radiosondes are important in securing the temporal homogeneity of climate data records (GCOS-134, 2009). This paper details results from comparison of GRUAN data products based on data collected by RS-11G and RS92 on dual flights conducted from Tateno between April 2015 to June 2017.





In this paper, Section 2 describes the instrumentation used and GRUAN data products (i.e., data processing details) for RS-11G and RS92, Section 3 outlines the methods used for dual launches, Section 4 details the comparison analysis methods, Section 5 gives the comparison results, Section 6 discusses outcomes from a dual flight of RS-11G and a cryogenic frostpoint hygrometer (CFH; Vömel et al., 2007, 2016), and Section 7 summarizes the findings.

## 2 Instrumentation

Table 1 shows the specifications of RS92 and RS-11G. The former is operated with a Vaisala DigiCora Sounding System III, and the latter is operated with a Meisei MGPS2.

### 2.1 Sensor material and GRUAN data processing for RS-11G

RS-11G has a thermistor temperature sensor and an electrostatic capacitance humidity sensor. Geopotential height is derived from GPS data, and pressure is derived from GPS geopotential height. Wind speed and wind direction are calculated from GPS Doppler speed data.

All the RS-11G radiosondes used are subjected to the manufacturer's specified ground check before launch. At this time,
the temperature and RH sensors are compared with reference sensors under indoor conditions (Appendix 7, Kizu et al., 2018). If the differences between RS-11G and the reference sensors are within $\Delta U < \pm 7\%RH$ and $\Delta T < \pm 0.5°C$, the radiosonde is considered suitable for observation. The units are also subjected to manufacturer-independent ground checking with a standard humidity chamber (SHC, Appendix F, Kizu et al., 2018) at least a day before launch. The SHC provides conditions of approximately 0%RH using a molecular sieve and 100%RH using a sponge saturated with distilled water. The
RH sensor of RS-11G is compared with the reference sensor for SHC during this additional ground check. The results of the SHC ground check are used to create new calibration coefficients together with the values of the original manufacturer's calibration, which is conducted between 15%RH and 95%RH (Kizu et al., 2018).

Figure 1 shows the processing flow followed to derive temperature measurement values from RS-11G GRUAN data processing-version 1. RS-11G observation data are collected at 1-second intervals and converted to create the RS-11G
GRUAN data product (Kizu et al., 2018). The received frequency for temperature is converted into thermistor resistance, which is then converted into temperature using factory-set calibration coefficients. The raw temperature data need to be corrected for heat spike errors and solar radiation errors. Heat spike errors result from air being heated by the sensor frame, package, and balloon, but warm air from the balloon may be the main source (e.g., Shimizu and Hasebe, 2010). Such errors are corrected when the string between the balloon and the radiosonde may be too short (e.g., 10 m with a 600 g balloon)
using minima-pass filtering and a moving-average procedure. Solar radiation errors result from solar heating, particularly at higher altitudes during daytime soundings. The amount of such heating can be theoretically estimated using a heat-balance





equation (JMA, 1995) as a function of solar radiative flux, solar elevation angle, pressure, temperature, and ventilation speed at the measurement time. Although there are other error sources such as infrared radiation, evaporative cooling when the thermistor is coated with water or ice during flight through a cloud layer, and sensor response time, correction to remove errors from these sources is not applied for the current RS-11G GDP because their impacts are negligible compared to the
above-mentioned sources or difficult to quantify.

Figure 2 shows the processing flow followed to derive RH measurement values for RS-11G GDP. The received frequency for RH is converted into capacitance, which is then converted into raw RH data using sensor-specific calibration coefficients. The raw RH data need to be corrected for sensor time-lag, contamination, temperature-humidity dependence, and sensor-versus-air temperature difference. The response time of thin-film polymer RH sensors increases exponentially at lower
temperatures, and has been measured in laboratory experiments at various temperature points in a chamber (Kizu et al., 2018). The response time also depends on the direction of change between wet and dry conditions. Current GRUAN data processing for RS-11G involves the use of response time values from dry to wet conditions because the use of values from wet to dry conditions could result in over-correction. A contamination filter is used to remove errors caused by water droplets or ice in rainy conditions. This type of wet contamination error manifests as spikes in the raw RH profile, and a
minimum filter is therefore applied to high-frequency components of raw RH data. The temperature-dependence of thin-film polymer RH sensors in colder environments was evaluated under laboratory conditions by comparison with reference values from a chilled mirror hygrometer, and a correction curve was developed using the least squares method. As the temperature of the RH sensor is not exactly the same as that of ambient air due to solar heating and heat conduction from the RS-11G unit, RH values from RS-11G need to be adjusted with respect to the saturation pressure of the ambient air temperature (a
process referred to as Ts/Ta correction). The temperature of the RH sensor is estimated using data on air temperature and the amount of solar heating on the RH sensor. A further error source is the hysteresis property of the RH sensor. The results of chamber experiments showed that RH values exhibited wet biases when the condition was changed from 100%RH to 0%RH. As related quantification is rather complicated, this influence is not corrected in the current GDP version.

Figure 3 shows the processing flow followed to derive geopotential height and pressure measurement values for RS-11G
GDP. Geopotential height is calculated from geometric altitude data provided by the GPS receiver on RS-11G. The offset between the altitude when the balloon is actually released and the altitude at release time, as determined by the sounding system, is added to the measurement value. As height data are also affected by the payload, a moving average is applied to the data with a 61-point-wide window.

Figure 4 shows the processing flow followed to derive horizontal wind measurement values for RS-11G GDP. Zonal and
meridional winds (U and V, respectively) are derived from GPS Doppler speed data. As U and V data include random noise caused by pendulum motion, as with height data, a low-pass digital filter with a Kaiser window (Appendix E, Kizu et al., 2018) is applied to remove this influence, and the final wind speed and wind direction data are derived from the smoothed values of U and V.

Further details of data processing for RS-11G GDP can be found in Kizu et al. (2018).



## 2.2 Sensor material and GRUAN data processing for RS92

RS92 has a capacitive wire temperature sensor, a thin-film capacitor with a heated twin humidity sensor, a silicon pressure sensor, and a GPS receiver (Dirksen et al., 2014). All RS92 units are subjected the manufacturer's specified ground check before launch. At this time, the temperature and RH sensors are inserted into a ground check unit (GC25) and heated to remove contamination. The temperature sensor is then compared with the reference sensors under indoor conditions and the RH sensor is checked under dry (about 0%RH) conditions maintained with a desiccant bed. Pressure is compared with the reference value of the Automated Meteorological Data Acquisition System (AMeDAS) run by JMA at Tateno. If the differences between RS92 and the reference values are within $\Delta U < \pm 4\%RH$, $\Delta T < \pm 1.0°C$, and $\Delta P < \pm 3.0$ hPa, the radiosonde is considered suitable for observation. Additional ground checking with the SHC (under 100%RH conditions) for RS92 is not conducted at Tateno station. Version 2 of RS92 GDP (Dirksen, 2014) was created at the GRUAN Lead Centre. Related processing is briefly outlined below.

The processing flow followed for temperature data is shown in Figure 2 of Dirksen et al. (2014). Raw temperature data are corrected for solar radiation errors and heat spike errors as with RS-11G. Solar radiation errors relate to overall direct and scattered solar irradiance, ambient pressure and ventilation, and are estimated from a radiative transfer model that takes into account the solar elevation angle at the measurement time. Vaisala radiation error correction data are also available in table form. GRUAN data processing for RS92 involves application of the average of the two, as it remains unclear which correction model is more appropriate (Dirksen et al., 2014, GRUAN-TD-4, 2016). Heat spike errors are removed by applying a low-pass digital filter with a cut-off frequency of 0.1 Hz (Dirksen et al., 2014).

RS92 RH sensors have a temperature-dependent dry bias. GRUAN data processing corrects for this based on multiplication with an empirical correction factor before other forms of correction are applied. The raw RH data are corrected for radiation dry bias, sensor time-lag, and temperature-dependence errors. Radiation dry bias is caused by solar heating on the RH sensors, and the same approach as for the temperature sensor is used to estimate the amount of correction required. RH sensor response slows at low temperatures, and time-lag becomes significant below −40°C. Time-lag is corrected based on the relationship between a time constant and temperature using a low-pass filter in the GRUAN data product for RS92 (Dirksen et al., 2014).

The RS92 used at Tateno has a pressure sensor and a GPS receiver, both of which can be used to calculate geopotential height. Pressure measurement data are used to derive geopotential height in the lower part of the profile where the signal-to noise performance of the pressure sensor is sufficient, and measurements from the GPS sensor are used in the upper part of the profile. The altitude of the switch is typically between 9 and 17 km (GRUAN-TD-4, 2016). The pressure sensor is recalibrated against the reference value from a station barometer during the ground check, and calculation is performed to determine the correction factor for application to the entire pressure profile during sounding (Dirksen et al., 2014).





U and V data are retrieved from the Doppler shift in the GPS carrier signal, and noise is removed using a low-pass digital filter. The smoothed data are converted into wind speed and direction values (Dirksen et al., 2014).

## 3 Methods used for dual launches

GDPs produced from RS-11G and RS92 data between April 2015 and June 2017 were chosen for this study. Among the 87 dual flights involved, 5 were excluded from the analysis due to significant RS-11G water/ice contamination issues, and 30 more were excluded because GDPs of the soundings were not made or there were in-flight instrumental issues. Analysis of 22 daytime (09 LT, 00 UTC) and 30 nighttime (21 LT, 12 UTC) measurements (52 in all) is reported below. Table 2 shows surface observations and balloon burst heights for each of the 52 flights. The burst heights were mostly above 30 km.

Figure 5 shows the flight configurations. For all soundings, a 1,200 g balloon was used. The RS-11G and RS92 units were attached to both ends of a 1 m or 0.9 m rod. Table 3 shows the details of the rigs used for the comparison flights. The bamboo rod used from April 2015 to September 2015 was replaced with a lightweight cardboard rod in October 2015 for safety in the event of a fall to the ground. During these periods, the radiosondes were directly attached to the rod with elevated temperature sensors to avoid any rod-related influence on temperature and humidity measurement. However, the

cardboard rod was thicker than the bamboo rod, which might have caused an unexpected influence on temperature measurement. Temperature differences with the cardboard rod tended to be somewhat larger than those with bamboo or plastic rods in the lower troposphere and the lower stratosphere, although the mean difference with the cardboard rod was essentially within the standard deviation of the mean difference for all soundings. Additionally, for radiosondes with a direct rod attachment, temperature differences can be expected due to varying sensor orientation with respect to the position of the

sun. Accordingly, the rig was replaced with a bamboo rod from which radiosondes were hung in September 2016. The latest rod, which is a plastic cardboard composite with an aluminum tape covering (Table 3) applied to reduce the effects of radiation, has been used since December 2016 based on the GRUAN recommendation (Rohden et al., 2016).

## 4 Comparison method

Data for the GDPs of RS-11G and RS92 are collected at 1-second intervals. The authors compared simultaneous observation data on the basis of time since balloon release using a statistical approach as per the approach adopted by Kobayashi et al. (2012) to evaluate differences in sensors and correction methods.

### 4.1 Time adjustment procedure

Observation data from each radiosonde have a time stamp from the relevant sounding system. As there may be minor



discrepancies in balloon-launch time stamps, these data are time-adjusted using temperature as a parameter based on Kobayashi et al. (2012). Values in any 5-minute period during which the temperature difference between two radiosondes is smaller than 1 K with a marked change (e.g., in the inversion layer) are chosen from temperature data between 3 and 30 minutes after balloon release. Correlation coefficients are calculated by shifting the two temperature profiles every second.

The lag time that gives the greatest correlation coefficient is used to shift one of the two sets of profiles. In this study, the time lag between RS-11G and RS92 was less than 3 seconds in most cases.

### 4.2 Statistical procedure

After time adjustment, per-second differences between RS-11G and RS92 measurements were calculated and the resulting data were allocated to the following 13 pressure layers based on RS92 pressure data ($P_i^{92}$, where 92 represents RS92 and i

indicates the time step)  as per Kobayashi et al. (2012):

1st layer : 1000 hPa $\geq P_i^{92} > 700$ hPa

2nd layer : 700 hPa $\geq$ $P_i^{92} > 500$ hPa

3rd layer : 500 hPa $\geq P_i^{92} > 300$ hPa

4th layer : 300 hPa $\geq P_i^{92} > 200$ hPa

5th layer : 200 hPa $\geq P_i^{92} > 150$ hPa

6th layer : 150 hPa $\geq P_i^{92} > 100$ hPa

7th layer : 100 hPa $\geq P_i^{92} > 70$ hPa

8th layer : 70 hPa $\geq P_i^{92} > 50$ hPa

9th layer : 50 hPa $\geq P_i^{92} > 30$ hPa

10th layer : 30 hPa $\geq P_i^{92} > 20$ hPa

11th layer : 20 hPa $\geq P_i^{92} > 15$ hPa

12th layer : 15 hPa $\geq P_i^{92} > 10$ hPa

13th layer : 10 hPa $\geq P_i^{92} > 5$ hPa

$A_i^{11G}$ and $A_i^{92}$ are RS-11G and RS92 values, respectively, at time step i. The mean of each variable ($\overline{A^{11G}}$, $\overline{A^{92}}$) and the mean

of the difference ($\overline{\Delta A}$) are calculated using Eqs. (1) – (3) below for each pressure layer (from i = is to i = ie). The difference is defined as the RS-11G value minus the RS92 value ($\Delta A_i = A_i^{11G} - A_i^{92}$).

$$\overline{A^{11G}} = \frac{\sum_{i=is}^{ie} A_i^{11G}}{ie - is + 1} \quad (1)$$

$$\overline{A^{92}} = \frac{\sum_{i=is}^{ie} A_i^{92}}{ie - is + 1} \quad (2)$$

$$\overline{\Delta A} = \frac{\sum_{i=is}^{ie} \Delta A_i}{ie - is + 1} \quad (3)$$



Statistics for each pressure layer are calculated separately for daytime, nighttime, and individual seasons. Figure 6 shows the number of flights for each season. M is defined as the total number of soundings (k=1,2,…,M) in each condition; e.g., M = 6 for daytime in spring, and M = 7 for nighttime in spring.

The ensemble mean of the RS-11G GDP for individual pressure layers with each condition is

$$\overline{\overline{A^{11G}}} = \frac{\sum_{k=1}^{M} \overline{A_k^{11G}}}{M} \quad (4)$$

The ensemble mean of RS92 GDP for each pressure layer for each condition is

$$\overline{\overline{A^{92}}} = \frac{\sum_{k=1}^{M} \overline{A_k^{92}}}{M} \quad (5)$$

The ensemble mean difference for each pressure layer is

$$\overline{\overline{\Delta A}} = \frac{\sum_{k=1}^{M} \overline{\Delta A_k}}{M} \quad (6)$$

The standard deviation of the ensemble mean difference for individual pressure layers for each condition is

$$\sigma = \sqrt{\frac{\sum_{k=1}^{M} \left( \overline{\Delta A_k} - \overline{\overline{\Delta A}} \right)^2}{M}} \quad (7)$$

Daytime observation is conducted at 00 UTC (9:00 LT, launched at 8:30 LT) and nighttime observation at 12 UTC (21:00 LT, launched at 20:30 LT). Spring is defined as March to May, summer June to August, autumn September to November, and
winter December to February. Figure 7 shows mean profiles of temperature and RH from RS-11G.

## 5 Results

### 5.1 Comparison of simultaneous measurements

Figure 8 shows ensemble mean temperature differences and the standard deviation of these differences. The RS92 GDP
was chosen as a reference in this study for its status as a GRUAN certified data product. In the stratosphere during the daytime, the RS-11G GDP value is about -0.4 K lower than the RS92 GDP value. At nighttime, temperature differences are very small at pressures > 20 hPa. Differences among the four seasons are limited.

Figure 9 shows ensemble mean RH differences and the standard deviation of these differences. The RH values of RS-11G GDPs are larger than those of RS92 GDPs, and the RH difference exceeds 2%RH between 500 and 150 hPa in both daytime
and nighttime data. Figure 10 shows ensemble mean RH differences classified for six RH ranges. Most samples in the 90–100%RH range are found at pressures > 300 hPa, and the RS-11G GDP value in this range is 2%RH smaller than the RS92 GDP value. The RH differences in the 50–70%RH and 70–90%RH ranges are very small at pressures > 500 hPa. In dry conditions with values less than 50%RH, the RS-11G GDP value is larger than the RS92 GDP value and the RH difference is approximately 5%RH between 500 and 150 hPa. RH differences in the 0–10%RH range are within 1%RH at pressures ≤ 70
25    hPa. The results shown in Fig. 10 also indicate that absolute RH differences at pressures > 500 hPa in Fig. 9 are smaller than



those between 500 and 200 hPa because the mean differences are balanced out by the values in both dry and humid conditions.

The RH sensor for the RS-11G GDP is checked using SHC values of 0%RH (a desiccant-based dry condition) and 100%RH (a distilled water-based wet condition) before launch, and the check data are utilized for correction of the RH

calibration curve in the GDP. Figure 11 shows RH profiles with and without SHC correction for 7 March 2016 at 12 UTC as an example of the effects of SHC correction. The difference between RS-11G GDP with correction and RS92 GDP is smaller in wet conditions at values greater than 90%RH, and SHC correction can therefore be deemed effective in this case. Half of the samples including very humid conditions indicate that SHC correction for RS-11G gives improved results. However, the effects of SHC correction for very dry conditions are relatively limited.

Figures 12 and 13 show seasonal ensemble mean differences of pressure and geopotential height and the related standard deviations, respectively. RS-11G GDP pressure is generally lower than that of RS92 GDP except at pressures > 700 hPa in summer and autumn. In the daytime, RS-11G GDP pressure is 0.5 hPa lower than RS92 GDP between 500 and 50 hPa, and the pressure difference is small at pressures ≤ 50 hPa. The pressure difference at nighttime is smaller than during the day. The measurement methods used contribute to pressure differences in tropospheric data; RS-11G GDP pressure is derived

from GPS data, while RS92 GDP pressure is derived from pressure sensor data. Temperature differences also influence pressure differences. In Fig. 8, the daytime temperature difference is larger than at nighttime, which may cause differences between daytime and nighttime data in pressure comparison results. The RS-11G GDP geopotential height is larger than that of RS92 GDP in the daytime, and the difference is 10–20 m at pressures ≤ 100 hPa. The geopotential height difference at nighttime is smaller than during the day. The daytime standard deviation in spring at pressures ≤ 30 hPa is much larger than

others, but if the exceptional case causing this is removed, the seasonal difference is very small.

Figure 14 shows wind speed and wind direction profiles from each RS-11G sounding. Figures 15, 16, 17, and 18 show seasonal ensemble mean differences of wind speed and wind direction. The mean wind speed differences are smaller than 0.2 m s$^{-1}$, and the mean wind direction differences are smaller than 1 degree. The mean wind component differences are also smaller than 0.1 m s$^{-1}$, and the standard deviations for all seasons are smaller than 0.1 m s$^{-1}$ between 700 and 15 hPa. As RS-

11G and RS92 both use GPS-based wind measurement procedures, RS-11G GDP winds and RS92 GDP winds show a close statistical correlation.

### 5.2 Case analysis with consideration of uncertainty estimates in GRUAN data products

An important aspect of GDP data is that uncertainty estimates are given for each measurement point to support climate

record quality. Immler et al. (2010) defined terminology for checking pairs of independent measurements of the same quantity for consistency using estimated uncertainties as described here. Consider two independent measurements, $m_1$ and $m_2$, of the same measurand with standard uncertainties $u_1$ and $u_2$, respectively. Assume that $m_1 = m_2$ is true and that uncertainty follows normal distribution. Expression of the degree of consistency between $m_1$ and $m_2$ is given as in Table 4,



where k is a coverage factor. Overall uncertainty is calculated from independent sources of uncertainties. The sources contributing to the RS-11G temperature and RH uncertainty budget are listed in Tables 5 and 6, respectively. Uncertainty estimates for RS92 and RS-11G GDPs are described in Dirksen et al. (2014) and Kizu et al. (2018), respectively.

Figure 19 shows temperature and RH profiles along with related uncertainties for a dual flight conducted at 00 UTC on 28
October 2016 as an example of a daytime flight. The radiosondes appear to pass through cloud layers around 850 and 500 hPa, at which a value of almost 100%RH is observed. The RS-11G GDP temperature is lower than that of RS92 GDP at pressures > 400 hPa, and the related difference is larger than the expanded uncertainty (with k = 2). The temperature difference is notably larger than the uncertainty when the RH drops quickly (around 850 and 500 hPa). The temperature difference between 400 hPa and the tropopause is within the standard uncertainty (with k = 1). For RH, the RS-11G GDP is
larger than the RS92 GDP after the radiosondes pass through the layer in which humidity drops rapidly (around 500 and 250 hPa) where the RH difference is larger than the expanded uncertainty. When radiosondes leave clouds, temperature data and RH sensors may be affected by cooling as water or ice evaporates from the sensor surface, leading to errors in measurement. Additionally, RH measurement may be affected by sensor hysteresis characteristics. The RS-11G GDP includes a noise filter that removes the influence of water or ice when radiosondes pass through clouds, and the RH sensor of RS92 has a heating
function to prevent icing during flight. The RS-11G GDP appears to be affected by water droplets or ice more than the RS92 GDP in this case. The temperature difference in the stratosphere is also larger than that in the troposphere, probably due to errors in the treatment of solar radiation effects for both GDPs. However, the difference is within the expanded uncertainty, and the discrepancy is categorized as being in agreement. The RH of the RS-11G GDP in the stratosphere is a few %RH degrees larger than the RS92 GDP and within the standard uncertainty. However, the water vapor mixing ratios derived from
the RH of the RS-11G GDP and the RS92 GDP in the lower stratosphere (between 70 and 60 hPa) in Fig. 19 are approximately 4.7 and 3.0 ppmv, respectively, and the difference between RS-11G GDP and RS92 GDP is around 1.7 ppmv, which is approximately half the RS92 GDP value. Hurst et al. (2016) reported that water vapor mixing ratios at 68 hPa in the northern middle latitudes are roughly 3.5–5.0 ppmv, and the mixing ratio discrepancy caused by differences in the measurement method (e.g., the difference between balloon-borne frost point hygrometers and Aura Microwave Limb
Sounders) may be 0.3 ppmv. Hurst et al. (2016) also reported that rates for the stratospheric average trend of the mixing ratio from 2010 to mid-2015 ranged from 0.03 to 0.07 ppmv per year[-1]. The RH difference between RS-11G GDP and RS92 GDP in Fig. 19 is much larger than these index values, and the RH sensors of RS-11G and RS92 are deemed too unresponsive for stratospheric evaluation in this case. Accordingly, no discussion will be made here regarding RH measurement differences in the stratosphere based on the results shown in Fig. 19.
Figure 20 shows a nighttime situation with a launch at 12 UTC on 4 November 2016. In contrast to the daytime situation, in which temperature uncertainty increases with height due to solar radiation, nighttime temperature uncertainty does not depend on height and remains within the standard range at pressures > 30 hPa. For RH, although RS-11G GDP is a few %RH degrees larger than RS92 GDP between 850 and 200 hPa at nighttime, the values correspond within the expanded uncertainty.





### 5.3 Consistency of temperature measurements from RS-11G and RS92

Overall uncertainty in GDPs is estimated from all sources of uncertainty, and measurement results can be assessed using the quantitated uncertainties of each source. The sources contributing to the RS-11G temperature uncertainty budget are

listed in Table 5 (Kizu et al., 2018). Uncertainty associated with filtering (including moving averaging) is derived using

$$\sqrt{\frac{N'}{N'-1}} \sum c_i^2 (T_{org} - T_{filtered}) \qquad (8)$$

$$N' = (\sum c_i^2)^{-1},$$

where $c_i$ represents the coefficients of filtering at time step i and $N'$ is the effective sample size. For the RS92 temperature, uncorrelated uncertainty is based on statistical uncertainty and determined via spike removal. Correlated uncertainty consists

of the uncertainty associated with radiation correction and the calibration uncertainty of the temperature sensor (Dirksen et al., 2014; Sommer et al., 2016). The standard uncertainty of each source for observation at 00 UTC on 28th October, 2016, is illustrated in Fig. 21. While sensor orientation derived from Table 2 in Dirksen et al. (2014) is the major source of uncertainty in RS92 temperature measurement, albedo is the major source for RS-11G because orientation is not explicitly considered in RS-11G GDPs. The JMA solar radiation correction model (JMA, 1995) assumes that the surface and cloud

albedo is constant at 20%. However, the actual albedo during the flight depends on surface and cloud conditions, and the correction amount is underestimated when highly reflective clouds are present (Kizu et al., 2018).

For statistical comparison, the percentages of consistency ranks (1, 2, 3 or 4) between RS92 GDP and RS-11G GDP in a particular pressure layer are calculated as follows:

a. Calculate the combined uncertainty $u_c$ for every data point with a 1-sec resolution.

$u_c = \sqrt{u_{92}^2 + u_{11G}^2} \qquad (9)$

b. Define $T_{11G}$, $T_{92}$ and $d = |T_{11G} - T_{92}|$ as temperature measurements from RS-11G GDP and RS92 GDP, and the absolute value of the difference between RS-11G GDP and RS92 GDP at every data point, respectively. Rank d is then defined as 1 ($d \leq u_c$: consistent), 2 ($u_c < d \leq 2u_c$: in agreement), 3 ($2u_c < d \leq 3u_c$: significantly different) and 4 ($3u_c < d$: inconsistent).

c. Arrange the rank values in ascending order for the pressure layer; the 95% value is assigned as the consistency rank of the layer for each flight.

The percentages of consistency ranking for all daytime and nighttime flights are illustrated in Fig. 22. While most measurements at pressures > 10 hPa are consistent at nighttime, most are not consistent during the daytime for all layers. Such uncertainty estimates enable vertical evaluation of measurement uncertainty.

### 5.4 Consistency of RH measurements from RS-11G and RS92



Sources contributing to the RS-11G RH uncertainty budget are listed in Table 6 (Kizu et al., 2018). The standard uncertainty of each source for observation at 00 UTC on 28th October, 2016, is illustrated in Fig. 23. For RS92 RH, uncertainty consists of correction for calibration uncertainty and temperature-dependent calibration uncertainty, radiation dry bias, the time-lag constant, and the statistical uncertainty of time-lag correction (Dirksen et al., 2014). As the calculation

method for each component with RS92 is not detailed in Dirksen et al. (2014), only uncorrelated and correlated uncertainties are illustrated for RS92. For RS-11G, the major source of uncertainty is temperature-humidity dependence correction for the whole layer. Statistical uncertainty and uncertainty from the sensor versus air temperature correction (green) are important in the lower and middle troposphere, and uncertainty from time-lag correction (red) is important near the tropopause.

The percentages of consistency ranking (calculated as per temperature) are illustrated in Fig. 24. In the middle troposphere,

half of RS92 GDP and RS-11G GDP values are significantly different or inconsistent. In the stratosphere, RS92 GDP and RS-11G GDP are always consistent. However, as discussed in Section 5.2, RH values in the stratosphere range across a few %RH degrees, and the RH sensors of RS92 and RS-11G are considered unresponsive in relation to conditions in the troposphere. Although the availability of measurement values from the stratosphere depends on the purpose of utilization and related accuracy requirements (Miloshevich et al., 2009), the consistency of RH measurements from the stratosphere is

not discussed here.

## 6 Comparison of RS-11G GDP humidity with CFH

At Tateno, radiosonde and Cryogenic Frostpoint Hygrometer (CFH, Vömel et al., 2007, 2016) comparison flights have been conducted twice a year since 2015. Figure 25 shows the results of a RS-11G and CFH comparison flight conducted on

10 November, 2016. This CFH is interfaced with RS-11G, and RH calculation for CFH involves the use of temperature values from RS-11G GDP. Figure 25 shows RH profiles from RS-11G GDP and CFH. The RH of the RS-11G GDP is around 7%RH greater than that of CFH between 500 and 200 hPa, and around 1%RH greater at pressures > 500 hPa. The difference is also more than 10%RH around 350 hPa, where RH drops rapidly and the difference is larger than the overall uncertainty. This significant difference is influenced by water or ice on the sensor and related hysteresis characteristics. The

tropopause is recorded at 100.5 hPa with temperatures lower than -70°C, and the RH difference is somewhat large, RS-11G GDP being a few percent smaller than CFH. In low-temperature conditions (Fig. 23), sensor time-lag and RS-11G RH sensor temperature dependence may be important factors in the humidity difference. Figure 26 shows results from a RS-11G, RS92, and CFH comparison flight conducted on 20 April, 2018. The RH of the RS-11G GDP is around 4%RH smaller than that of CFH at pressure > 700 hPa, while that of the RS92 GDP is largely in agreement with the RH of CFH. Meanwhile, the RH of

the RS-11G GDP is 2%RH greater than that of CFH between 400 and 200 hPa, while the RH of the RS92 GDP is 2%RH smaller than that of CFH. These comparisons indicate that the RS-11G GDP has a wet bias between 400 and 200 hPa, as is common in the results shown in Fig. 9.



## 7 Summary

To help characterize the GDPs of RS-11G and RS92, data collected on dual flights conducted from Tateno between April 2015 and June 2017 were analyzed in this study. The RS-11G GDP temperature was around -0.4 K lower than RS92 GDP data in daytime measurement in the stratosphere, while nighttime measurements generally corresponded. The RS-11G GDP

RH was 2%RH smaller than the RS92 GDP for 90–100%RH, and the RS-11G GDP was around 5%RH larger than the RS92 GDP at values lower than 50%RH. The effects of SHC correction were also verified, with samples featuring highly humid conditions showing improved results for RS-11G data. The pressure difference was 0.5 hPa in the troposphere, and the geopotential height difference was around 10–20 m in the stratosphere.

The consistency of temperature and RH measurements from RS-11G and RS92 with uncertainties was also analyzed. The

major sources of uncertainty in temperature measurements for RS-11G and RS92 GDPs were albedo and sensor orientation, respectively. Statistical comparison showed that most daytime temperature measurements were not consistent for any pressure layer. For RH measurements, the major source of uncertainty for the RS-11G GDP was temperature-humidity dependence correction for the whole layer, and half of RS92 and RS-11G GDP values were significantly different or inconsistent in the middle troposphere.

RS-11G GDP RH data were also evaluated based on comparison with CFH data, with results showing a wet bias in the former from CFH values between 400 and 200 hPa. The same characteristic was also observed in comparison with RS92 GDP data. The extent of CFH measurements to date remains limited, but plans are being made to conduct temperature and humidity measurement using high-quality radiosondes twice a year along with continuous comparison flights of a high-quality radiosonde and a routine radiosonde to facilitate GDP evaluation and further analysis of RS-11G characteristics.

This study involved evaluation of the characteristics of RS-11G GDP values with RS92 GDP as base data due to the latter's GRUAN radiosonde certification. The GRUAN certification process for RS-11G is underway, and ongoing analysis of GDP data is considered important for the provision of high-quality products to the user community.

Acknowledgments. The authors are grateful to the Aerological Observatory, the Observation Division of JMA's Observation

Department and Meisei Electric for their support and helpful advice. Thanks are also due to GRUAN staff for their support.

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





**Table 1:** Specifications of radiosondes and GRUAN data products (Meisei, 2016; Valsala, 2013; Dirksen et al., 2014, Kizu et al., 2018).

| Radiosonde | | RS-11G (RS-11G GPSsonde) | RS92 (RS92-SGP GPSsonde) |
|---|---|---|---|
| Temperature | Sensor | Type: Thermistor<br>Range: -90°C to +60°C<br>Resolution: 0.1°C | Type: Capacitive wire<br>Range: -90°C to +60°C<br>Resolution: 0.1°C |
| | Correction procudures in the GDP* | Heat spike filtering<br>Solar radiation correction | Heat spike filtering<br>Solar radiation correction (average of GRUAN and Vaisala correction model) |
| Humidity | Sensor | Type: Electrostatic capacitance humidity sensor<br>Range: 0%RH to 100%RH<br>Resolution: 0.1%RH | Type: Thin-film capacitor, Headed twin sensor<br>Range: 0%RH to 100%RH<br>Resolution: 1%RH |
| | Saturation vapour pressure formulation | Hyland and Wexler equation | Hyland and Wexler equation (Hyland and Wexler, 1983) |
| | Correction procudures in the GDP | Contamination removal filter for rain and cloud droplets<br>Time-lag correction<br>Temperature-Humidity dependent bias correction<br>Sensor versus air temperature correction | Radiation dry bias correction<br>Time-lag correction<br>Temperature dependent bias correction |
| Pressure/ Geopotential height | Sensor | Type: GPS<br>Range: 1050.0 hPa to 3.0 hPa<br>Resolution: 0.1 hPa | Type: Silicon pressure sensor, and GPS<br>Range: 1080 hPa to 3 hPa<br>Resolution: 0.1 hPa |
| | Calculation | Pressure is calculated from the GPS geopotential height using the hypsometric equation | In the lower part of the profile: the pressure sensor is used, and the geopotential height is derived from pressure using the hypsometric equation<br>In the upper part of the profile: use the GPS sensor |
| | Correction procedures in the GDP | Offset between the balloon release altitude and the altitude at the release time | The pressure sensor is recalibrated against the station barometer |
| Wind | Correction procedures in the GDP | GPS wind finding (with SBAS)<br>Smoothing (a low-pass digital filter is used to remove noises) | GPS wind finding (with GBAS)<br>Smoothing (a low-pass digital filter is used to remove noises) |
| Dimensions(DWH) | | 67 × 86 × 155 (mm) | 75 × 80 × 220 (mm) |
| Weight | | 85 g (with a battery) | 290 g (with batteries) |
| Ground System | | MGPS2 (Version 2) | Vaisala DigiCORA III (Version 3.64) |
| Photo | | ![RS-11G photo] | ![RS92 photo] |

*: GRUAN data product





**Table 2:** Flight information including surface observation, cloud cover data, and balloon burst height from RS-11G data

| Flight Number | Date | Time (LST) | Pressure (hPa) | Temperature (°C) | Humidity (%RH) | Wind Direction (degree) | Wind Speed (ms⁻¹) | N | Nh | CL | h | CM | CH | Weather | Height (km) | Pressure (hPa) |
|---|---|---|---|---|---|---|---|---|---|---|---|---|---|---|---|---|
| 1 | 4/21/2015 | 20:30:17 | 1013.2 | 12.4 | 71 | 90 | 2.9 | 7 | 7 | 0 | / | 7 | / | 02 | 36.006 | 4.8 |
| 2 | 5/7/2015 | 20:30:20 | 1008.9 | 15.5 | 77 | 130 | 1.7 | 1 | 0 | 0 | 9 | 0 | 2 | 02 | 35.166 | 5.5 |
| 3 | 5/11/2015 | 8:30:21 | 1020.1 | 15.9 | 33 | 30 | 1.5 | 1 | 0 | 0 | 9 | 0 | 1 | 02 | 34.285 | 6.2 |
| 4 | 5/25/2015 | 8:30:15 | 1008.0 | 21.0 | 68 | 70 | 3.4 | 6 | 1 | 2 | / | 3 | 2 | 02 | 34.467 | 6.2 |
| 5 | 6/1/2015 | 20:30:14 | 1011.0 | 21.0 | 74 | 140 | 2.1 | 7 | 0 | 0 | 9 | 0 | 2 | 02 | 36.531 | 4.7 |
| 6 | 6/16/2015 | 20:30:14 | 1002.4 | 22.0 | 89 | 80 | 2.4 | 8 | 3 | 5 | / | 7 | / | 02 | 36.961 | 4.5 |
| 7 | 6/22/2015 | 8:30:16 | 1010.0 | 22.1 | 78 | 40 | 1.3 | 7 | 6 | 2 | / | 3 | 2 | 02 | 34.917 | 6.0 |
| 8 | 9/28/2015 | 8:30:17 | 1013.1 | 22.8 | 72 | 310 | 1.0 | 0 | 0 | 0 | 9 | 0 | 0 | 02 | 35.320 | 5.4 |
| 9 | 10/5/2015 | 20:30:12 | 1019.6 | 15.8 | 68 | 50 | 2.2 | 7 | 1 | 0 | 9 | 3 | 2 | 02 | 34.366 | 6.2 |
| 10 | 10/13/2015 | 8:30:15 | 1010.3 | 18.0 | 73 | 300 | 1.3 | 0 | 0 | 0 | 9 | 0 | 0 | 02 | 33.581 | 6.9 |
| 11 | 10/22/2015 | 20:30:16 | 1017.6 | 14.8 | 80 | 40 | 2.4 | 7 | 3 | 5 | / | 3 | / | 02 | 34.373 | 6.2 |
| 12 | 10/26/2015 | 8:30:15 | 1017.7 | 12.4 | 55 | 260 | 0.6 | 0 | 0 | 0 | 9 | 0 | 0 | 02 | 34.078 | 6.3 |
| 13 | 11/30/2015 | 20:30:16 | 1016.6 | 7.2 | 95 | 300 | 2.0 | 1 | 0 | 0 | 9 | 0 | 2 | 01 | 34.664 | 6.0 |
| 14 | 12/7/2015 | 20:30:14 | 1018.9 | 10.6 | 72 | 40 | 2.9 | 7 | 7 | 5 | / | 7 | / | 02 | 35.327 | 5.5 |
| 15 | 12/21/2015 | 8:32:15 | 1021.1 | 4.5 | 73 | 290 | 1.6 | 7 | 7 | 0 | / | 7 | / | 02 | 34.757 | 6.0 |
| 16 | 12/28/2015 | 20:30:16 | 1017.7 | 0.6 | 69 | 90 | 1.0 | 1 | 1 | 0 | / | 3 | 0 | 02 | 30.301 | 11.4 |
| 17 | 1/4/2016 | 8:30:17 | 1012.4 | 4.5 | 88 | 280 | 0.6 | 0 | 0 | 0 | 9 | 0 | 0 | 02 | 33.615 | 6.7 |
| 18 | 1/12/2016 | 20:30:14 | 1011.1 | -0.6 | 93 | 310 | 1.3 | 0 | 0 | 0 | 9 | 0 | 0 | 02 | 34.548 | 5.8 |
| 19 | 1/25/2016 | 20:30:15 | 1018.3 | -1.7 | 54 | 150 | 2.4 | 1 | 1 | 1 | / | 0 | 0 | 02 | 34.099 | 6.1 |
| 20 | 2/8/2016 | 20:30:15 | 1013.2 | 0.7 | 80 | 40 | 1.6 | 1 | 1 | 1 | / | 0 | 0 | 02 | 34.079 | 6.3 |
| 21 | 2/15/2016 | 8:30:17 | 998.3 | 9.7 | 64 | 70 | 2.3 | 8 | 8 | 5 | / | / | / | 02 | 33.586 | 6.6 |
| 22 | 3/7/2016 | 20:30:15 | 1013.8 | 13.6 | 99 | 50 | 1.3 | 6 | 6 | 0 | / | 3 | 0 | 10 | 33.188 | 6.9 |
| 23 | 3/22/2016 | 20:30:19 | 1008.9 | 9.1 | 84 | 120 | 1.6 | 0 | 0 | 0 | 9 | 0 | 0 | 02 | 34.235 | 6.0 |
| 24 | 4/1/2016 | 8:30:16 | 1017.5 | 13.7 | 44 | 60 | 3.4 | 3 | 1 | 8 | / | 0 | 1 | 02 | 34.147 | 6.1 |
| 25 | 4/29/2016 | 8:30:18 | 994.4 | 17.0 | 41 | 310 | 5.9 | 5 | 2 | 8 | / | 3 | 2 | 02 | 33.746 | 6.7 |
| 26 | 6/3/2016 | 20:30:16 | 1011.3 | 18.8 | 62 | 150 | 2.7 | 0 | 0 | 0 | 9 | 0 | 0 | 02 | 35.920 | 5.1 |
| 27 | 6/17/2016 | 20:30:13 | 1003.9 | 23.3 | 85 | 70 | 1.8 | 7 | 7 | 2 | / | / | / | 02 | 26.471 | 20.9 |
| 28 | 9/30/2016 | 8:30:14 | 1016.3 | 18.2 | 63 | 50 | 2.9 | 8 | 8 | 5 | / | / | / | 02 | 34.619 | 5.9 |
| 29 | 10/14/2016 | 8:30:14 | 1023.0 | 15.0 | 68 | 360 | 2.0 | 7 | 6 | 0 | / | 3 | 2 | 02 | 31.569 | 9.3 |
| 30 | 10/28/2016 | 8:32:12 | 1018.4 | 12.5 | 77 | 360 | 1.2 | 8 | 8 | 5 | / | / | / | 02 | 35.435 | 5.1 |
| 31 | 11/4/2016 | 20:30:14 | 1014.1 | 9.5 | 81 | 90 | 1.3 | 7 | 7 | 0 | / | 3 | / | 02 | 35.303 | 5.1 |
| 32 | 11/18/2016 | 20:30:13 | 1021.0 | 7.6 | 89 | 300 | 1.2 | 7 | 7 | 5 | / | 0 | 0 | 10 | 35.651 | 4.8 |
| 33 | 11/25/2016 | 8:37:12 | 1019.7 | 1.2 | 93 | 310 | 1.3 | 5 | 2 | 0 | / | 3 | 2 | 02 | 20.566 | 48.8 |
| 34 | 12/2/2016 | 20:30:14 | 1020.5 | 5.7 | 68 | 60 | 1.8 | 1 | 0 | 0 | 9 | 0 | 2 | 02 | 35.665 | 5.1 |
| 35 | 12/9/2016 | 8:30:15 | 1009.9 | 5.8 | 60 | 310 | 2.0 | 0 | 0 | 0 | 9 | 0 | 0 | 02 | 36.896 | 4.4 |
| 36 | 12/16/2016 | 20:30:14 | 1013.5 | 2.1 | 37 | 270 | 3.0 | 0 | 0 | 0 | 9 | 0 | 0 | 02 | 35.827 | 4.9 |
| 37 | 12/23/2016 | 8:32:16 | 994.9 | 12.7 | 94 | 260 | 2.2 | 6 | 6 | 8 | / | 0 | 0 | 10 | 34.979 | 5.5 |
| 38 | 12/30/2016 | 20:30:14 | 1022.3 | -0.9 | 64 | 300 | 0.8 | 0 | 0 | 0 | 9 | 0 | 0 | 02 | 34.883 | 5.7 |
| 39 | 1/6/2017 | 8:40:16 | 1022.8 | 1.7 | 55 | 340 | 0.5 | 7 | 0 | 0 | 9 | 0 | 2 | 02 | 28.553 | 14.2 |
| 40 | 1/13/2017 | 20:30:14 | 999.0 | 1.9 | 73 | 100 | 1.7 | 2 | 2 | 2 | / | 0 | 0 | 02 | 26.877 | 18.2 |
| 41 | 1/27/2017 | 20:30:15 | 1014.1 | 7.6 | 50 | 300 | 1.7 | 1 | 1 | 1 | / | 3 | 0 | 02 | 32.090 | 8.2 |
| 42 | 2/10/2017 | 20:30:12 | 1000.2 | 2.5 | 34 | 360 | 2.1 | 7 | 0 | 0 | 9 | 0 | 2 | 02 | 32.346 | 7.8 |
| 43 | 2/24/2017 | 20:30:13 | 1013.2 | 4.5 | 33 | 50 | 1.1 | 8 | 8 | 8 | / | / | / | 02 | 34.834 | 5.3 |
| 44 | 3/3/2017 | 8:30:15 | 1005.0 | 6.8 | 86 | 280 | 1.8 | 0 | 0 | 0 | 9 | 0 | 0 | 02 | 35.128 | 5.1 |
| 45 | 3/10/2017 | 20:30:14 | 1010.9 | 5.9 | 32 | 300 | 2.1 | 2 | 1 | 1 | / | 3 | 0 | 02 | 32.679 | 7.5 |
| 46 | 3/24/2017 | 20:30:12 | 1015.7 | 3.6 | 48 | 320 | 1.8 | 1 | 1 | 1 | / | 0 | 0 | 02 | 33.889 | 6.4 |
| 47 | 4/7/2017 | 20:30:09 | 1013.3 | 16.5 | 92 | 160 | 2.7 | 6 | 2 | 8 | / | 0 | 2 | 02 | 32.877 | 7.5 |
| 48 | 4/14/2017 | 8:35:14 | 1015.8 | 13.1 | 44 | 280 | 1.9 | 0 | 0 | 0 | 9 | 0 | 0 | 02 | 30.477 | 10.8 |
| 49 | 6/2/2017 | 20:30:15 | 995.1 | 18.6 | 45 | 300 | 3.1 | 0 | 0 | 0 | 9 | 0 | 0 | 02 | 35.107 | 5.8 |
| 50 | 6/9/2017 | 8:34:15 | 1005.6 | 21.7 | 71 | 100 | 3.5 | 7 | 4 | 2 | / | 7 | / | 02 | 35.364 | 5.6 |
| 51 | 6/16/2017 | 20:30:14 | 1005.8 | 18.5 | 91 | 40 | 2.4 | 7 | 7 | 0 | / | 7 | / | 02 | 33.281 | 7.6 |
| 52 | 6/23/2017 | 8:30:13 | 1005.9 | 24.6 | 68 | 240 | 0.6 | 2 | 1 | 1 | / | 0 | 2 | 02 | 34.476 | 6.4 |





**Table 3:** Rig configurations adopted for dual observation

| Rig | Bamboo rod. The sensor booms are fixed and are pointing to opposite directions. | Cardboard rod. The sensor booms are fixed and are pointing to the outward direction. | Bamboo rod. Sondes are hanging and rotating freely. | Plastic cardboard rod with aluminum tape. Sondes are hanging and rotating freely. |
|---|---|---|---|---|
| Photo | | | | |
| Length of the rod | 1 m | 1 m | 1 m | 0.9 m |
| Period | April 2015– September 2015 | October 2015– June 2016 | September 2016– November 2016 | December 2016– |

5 **Table 4:** Terminology for comparing pairs of independent measurements of the same quantification for consistency; excerpt from Section 2 of Immler et al. (2010)

| $\lvert m_1 - m_2 \rvert < k\sqrt{u_1^2 + u_2^2}$ | TRUE | FALSE | Significant level |
|---|---|---|---|
| k=1 | consistent | suspicious | 32% |
| k=2 | in agreement | significantly different | 4.5% |
| k=3 | - | inconsistent | 0.27% |



**Table 5:** Sources contributing to RS-11G temperature measurement uncertainty

| Source | Description | Value |
|---|---|---|
| Calibration of T sensor $u_{Tcalib1}$ | Provided by the manufacturer | $0.3/\sqrt{3}$ |
| Variation of Temperature in calibration chamber $u_{Tcalib2}$ | Provided by the manufacturer | $0.13/\sqrt{3}$ |
| Averaging (filtering) $u_{std}(T)$ | Derived by Eq. (8) | Depending on the measurement |
| Albedo $u_{albedo}(T)$ | $\frac{1}{2\sqrt{3}}\lvert T_{cor}(albedo=0.6) - T_{cor}(albedo=0.1)\rvert$ | Depending on the measurement |
| Ventilation $u_{ventilation}(T)$ | $\frac{1}{\sqrt{3}}\lvert T_{cor}(v=asc+u(v)) - T_{cor}(v=asc)\rvert$ <br> $u(v)=3.0$ | Depending on the measurement |
| Correlated $u_{cor}(T)$ | $\sqrt{u_{Tcalib1}^2 + u_{Tcalib2}^2 + u_{albedo}^2(T) + u_{ventilation}^2(T)}$ | Depending on the measurement |
| Total $u(T)$ | $\sqrt{u_{cor}^2(T) + u_{std}^2(T)}$ | |

5    **Table 6:** Sources contributing to RS-11G RH measurement uncertainty

| Source | Description | Value |
|---|---|---|
| Calibration of RH sensor $u_{Ucalib}$ | Provided by the manufacturer | $2/\sqrt{3}$ |
| Frequency splitting $u_{std1}(U)$ | Derived by Eq. (8) | Depending on the measurement |
| Contamination correction $u_{std3}(U)$ | Derived by Eq. (8) | Depending on the measurement |
| Moving averaging $u_{std4}(U)$ | Derived by Eq. (8) | Depending on the measurement |
| Time lag correction $u_{TL}(U)$ | $\frac{1}{2\sqrt{3}}\lvert u(\tau=\tau_U+u(\tau_U)) - U(\tau=\tau_U+u(\tau_U))\rvert$ <br> $u(\tau_U)=0.25$ | Depending on the measurement |
| TUD correction $u_{TUD}(U)$ | | 1.8 |
| Ts/Ta correction $u_{T_s}(U)$ | $\frac{1}{2\sqrt{3}}\lvert U(T_s=T_{s,fin}+u(T_s)$ <br> $- U(T_s=T_{s,fin}-u(T_s)\rvert$ <br> $u(T_s)=0.3$ | Depending on the measurement |
| Hysteresis $u_{hysteresis}(U)$ | only when relative humidity is decreasing | 0 ($\Delta U/\Delta t \geq -0.05$) <br> $1.8/\sqrt{3}$ ($\Delta U/\Delta t < -0.05$) |
| Statistical $u_{std}(U)$ | $\sqrt{u_{std1}^2(U) + u_{std3}^2(U) + u_{std4}^2(U)}$ | Depending on the measurement |
| Correlated $u_{cor}(U)$ | $\sqrt{u_{Ucalib}^2 + u_{TL}^2(U) + u_{TUD}^2(U) + u_{T_s}^2(U) + u_{hysteresis}^2(U)}$ | Depending on the measurement |
| Total $u(U)$ | $\sqrt{u_{std}^2(U) + u_{cor}^2(U)}$ | |



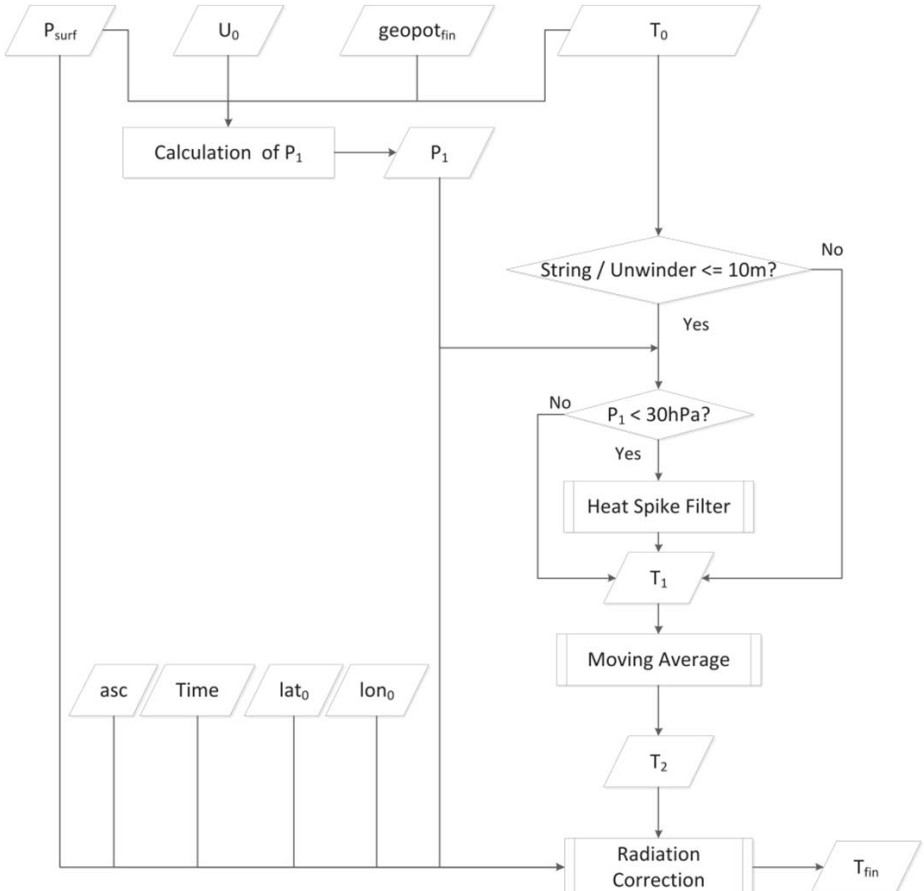

**Figure 1:** GRUAN data processing flow for RS-11G temperature measurement (excerpt from Kizu et al., 2018). $T_0$ and $U_0$ represent uncorrected temperature and RH, respectively, $P_{surf}$ is surface pressure, $lat_0$ and $lon_0$ are the initial data set of GPS latitude and longitude, respectively, geopot$_{fin}$ is the final geopotential height as derived from GPS altitude and latitude, asc is the ascent rate, and $T_{fin}$ is the corrected final temperature value.



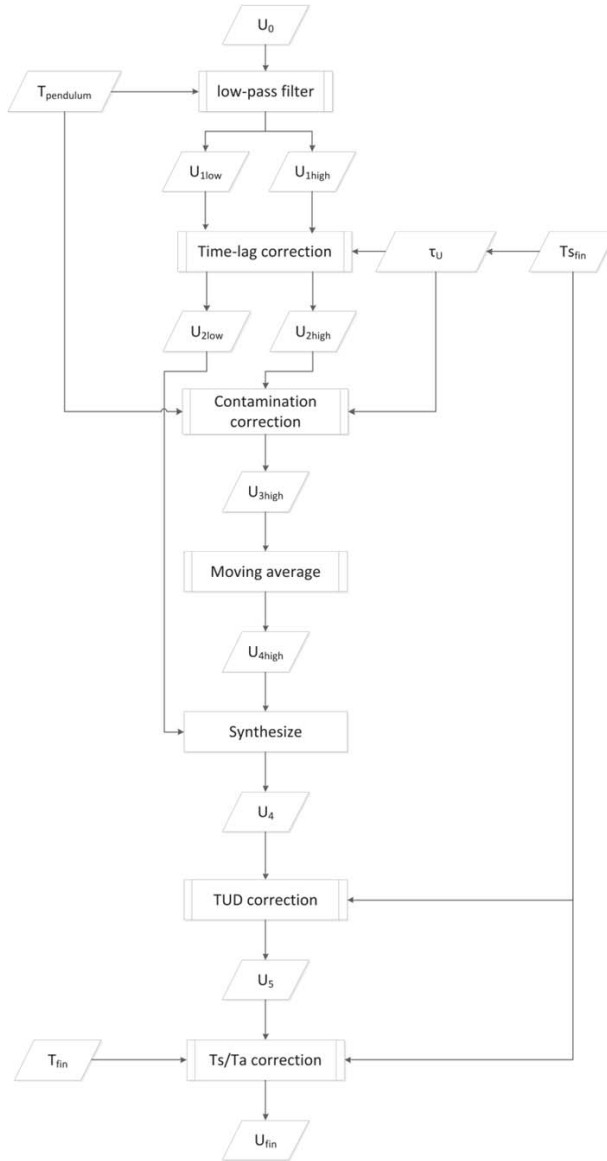

**Figure 2:** GRUAN data processing flow for RS-11G RH measurement (Kizu et al., 2018). $U_0$ is the uncorrected RH, $T_{pendulum}$ is the period of pendulum motion, $U_{1low}$ and $U_{1high}$ are low- and high-frequency components of $U_0$, $\tau_U$ is the sensor response time, $Ts_{fin}$ is the corrected RH sensor temperature, $T_{fin}$ is the corrected final temperature, and $U_{fin}$ is the corrected final RH value.





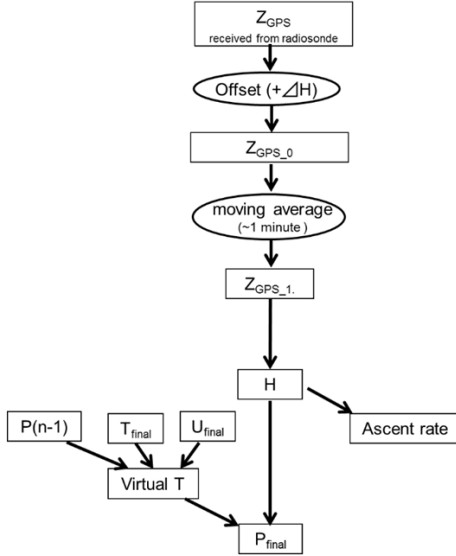

**Figure 3:** GRUAN data processing flow for RS-11G geopotential height and pressure measurement (Kizu et al., 2018). $Z_{GPS}$ is the GPS geometric altitude, $\Delta H$ is the offset between balloon release altitude and GPS geometric altitude upon balloon release, H is the geopotential height, $T_{final}$ and $U_{final}$ are the corrected final temperature and RH, respectively, and $P_{final}$ is the corrected final pressure.

**Figure 4:** GRUAN data processing flow for RS-11G horizontal wind measurement (Kizu et al., 2018). $U_0$ and $V_0$ are uncorrected zonal wind and meridional wind, respectively, $U_1$ is smoothed zonal wind, and $V_1$ is smoothed meridional wind.





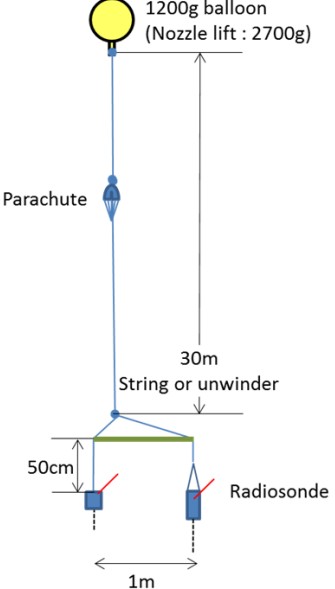

**Figure 5:** Flight configuration

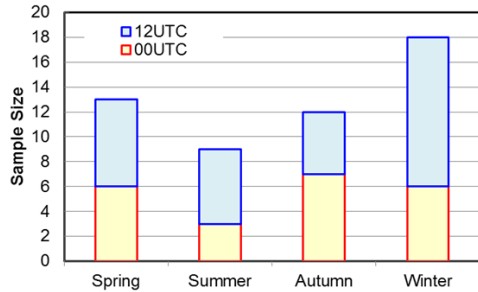

**Figure 6:** Number of samples for each season and for daytime/nighttime



**Figure 7:** Seasonal profiles of temperature and RH from RS-11G. Red and blue lines show daytime and nighttime observations, respectively, and black lines show means for all observations.





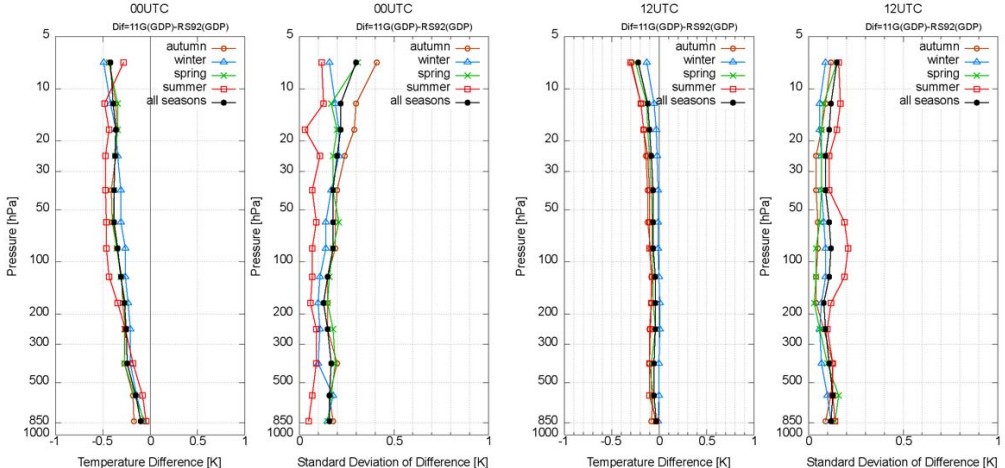

**Figure 8:** Profiles of mean temperature differences (RS-11G GDP minus RS92 GDP) and standard deviations for each season and for all seasons combined. Brown, blue, green, and red lines show means for autumn, winter, spring, and summer, respectively, and black lines show means for all seasons.

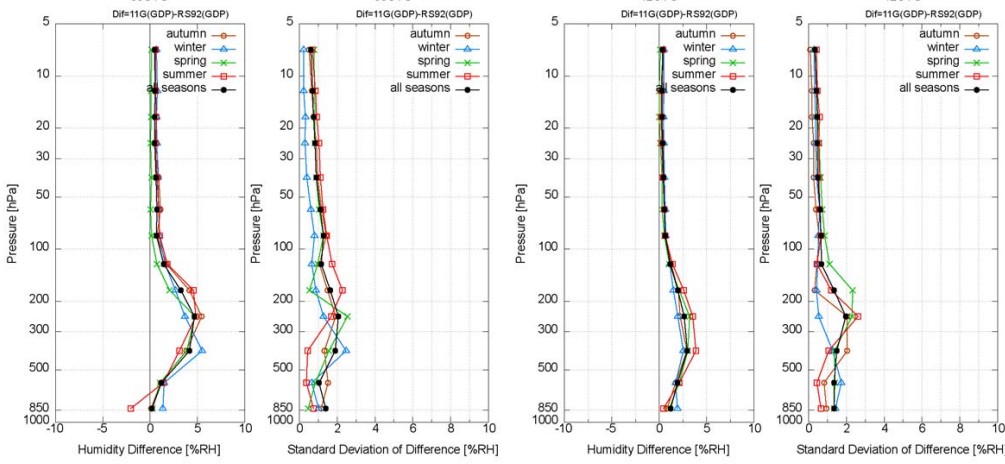

**Figure 9:** As per Fig. 8, but for RH





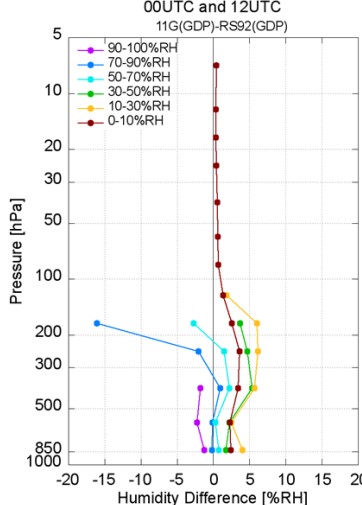

**Figure 10:** Profiles of mean differences (RS-11G GDP minus RS92 GDP) for all seasons and day and night combined for different RH ranges

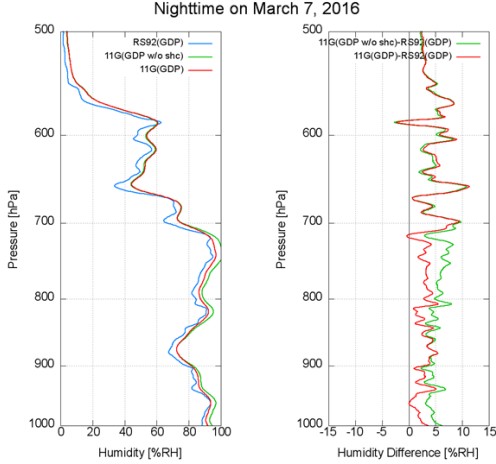

**Figure 11:** Profiles of relative humidity and differences (RS-11G GDP minus RS92 GDP). RS-11G (GDP) shows RS-11G GDP RH with SHC correction, and RS-11G (GDP w/o SHC) shows RS-11G GDP RH without SHC correction. RS92 data are not included in SHC checking at Tateno.



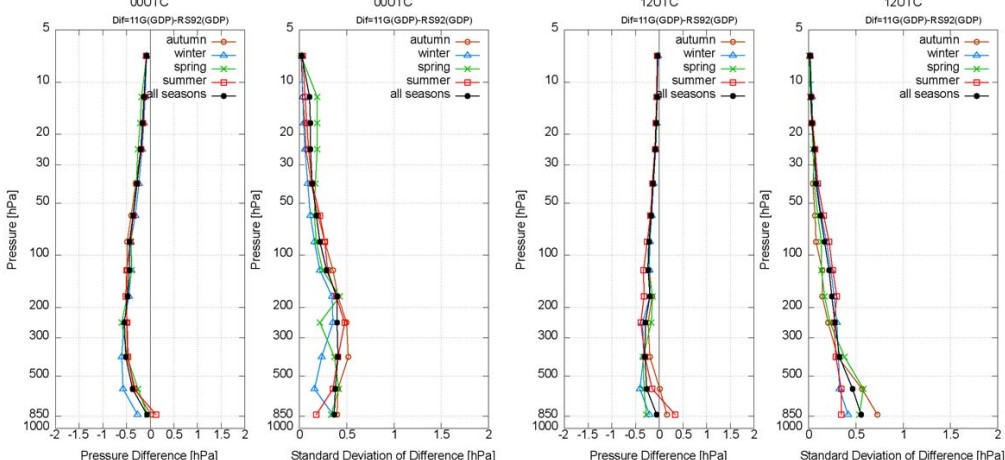

**Figure 12:** As per Fig. 8, but for pressure

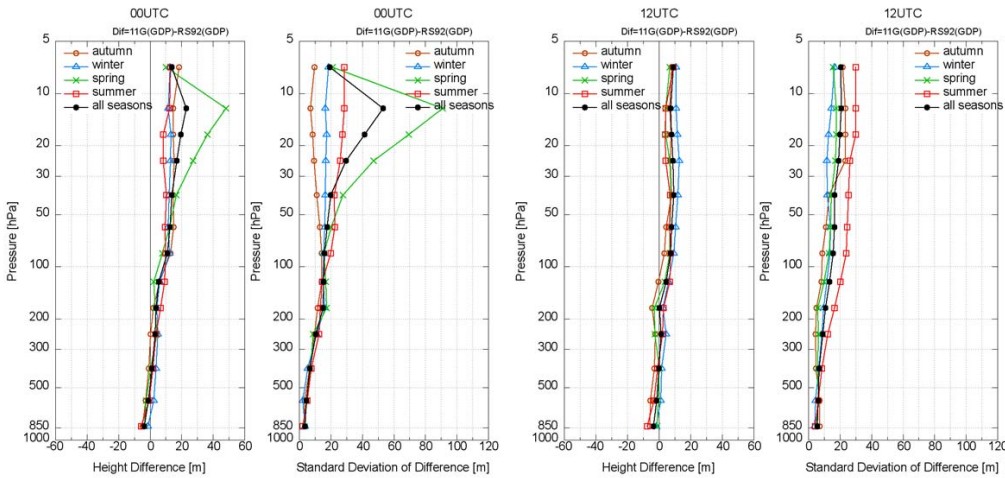

**Figure 13:** As per Fig. 8, but for geopotential height







**Figure 14:** As per Fig. 7, but for wind speed and direction



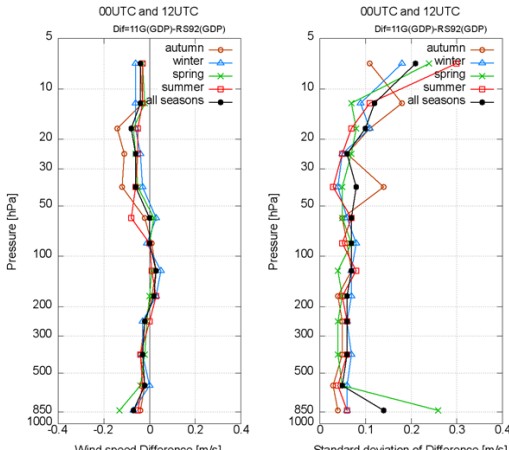

**Figure 15:** As per Fig. 8, but for wind speed for all soundings (00 and 12 UTC combined)

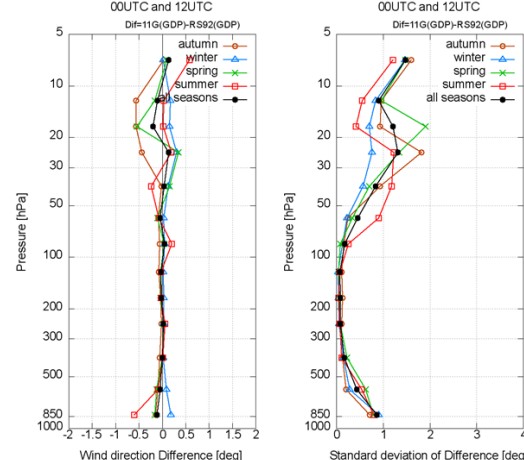

**Figure 16:** As per Fig. 15, but for wind direction


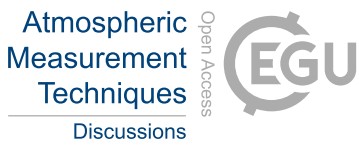

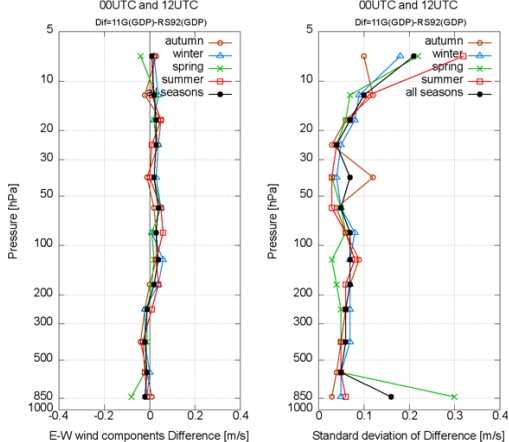

**Figure 17:** As per Fig. 15, but for U

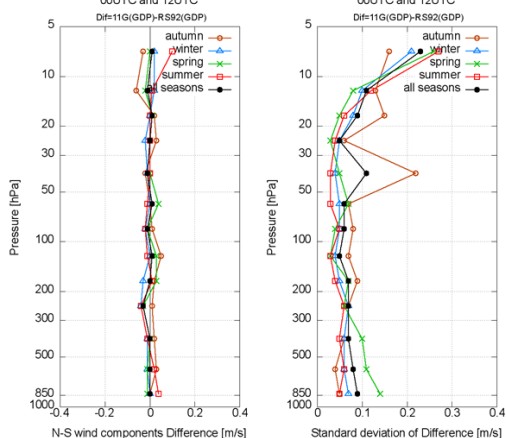

5  **Figure 18:** As per Fig. 15, but for V



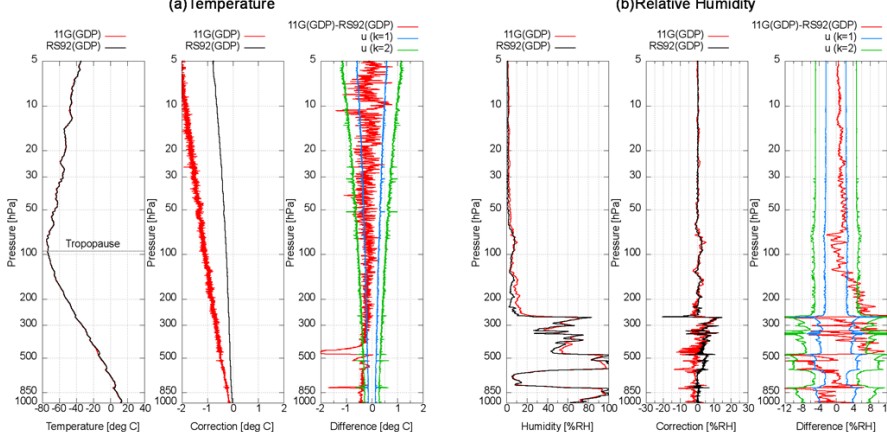

**Figure 19:** Temperature (a) and RH (b) profiles from the dual flight launched at 8:32 (LST) on 28th October, 2016. The panels on the left in (a) and (b) show temperature and RH profiles, respectively, from RS-11G GDP (red) and RS92 GDP (black). The middle panels show the total amount of correction. The panels on the right show differences (RS-11G GDP minus RS92 GDP) (red) and estimated uncertainties for k = 1 in blue and k = 2 in green. RH results for the stratosphere are not discussed here because measurement values for the stratosphere are considered to exceed the limit for reliable measurement.

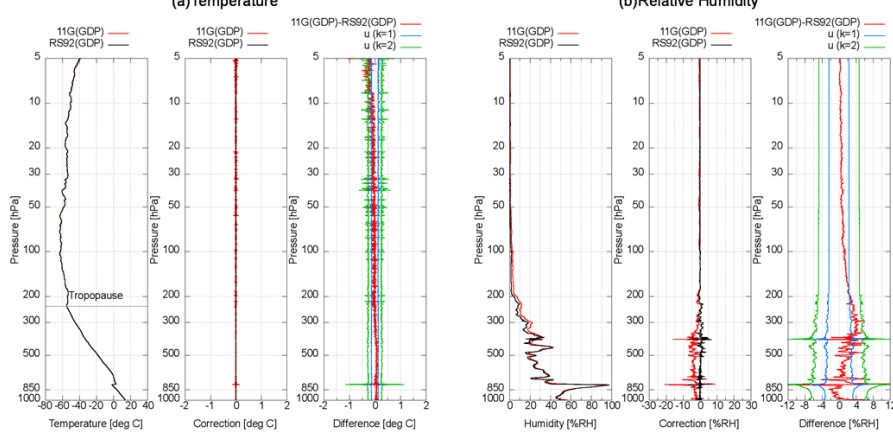

**Figure 20:** As per Fig. 19 but for the flight launched at 20:30 (LST) on 4th November, 2016.





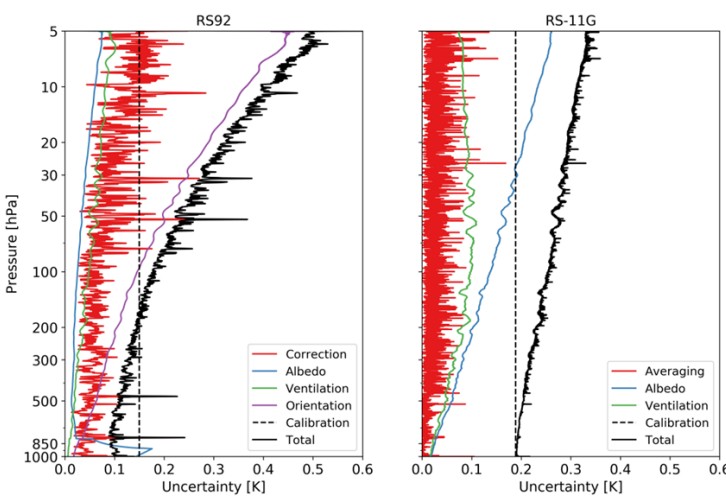

**Figure 21:** Uncertainty budget for temperature measurements at 00 UTC (daytime) on 28th October, 2016, for RS92 GDP (left) and RS-11G GDP (right)

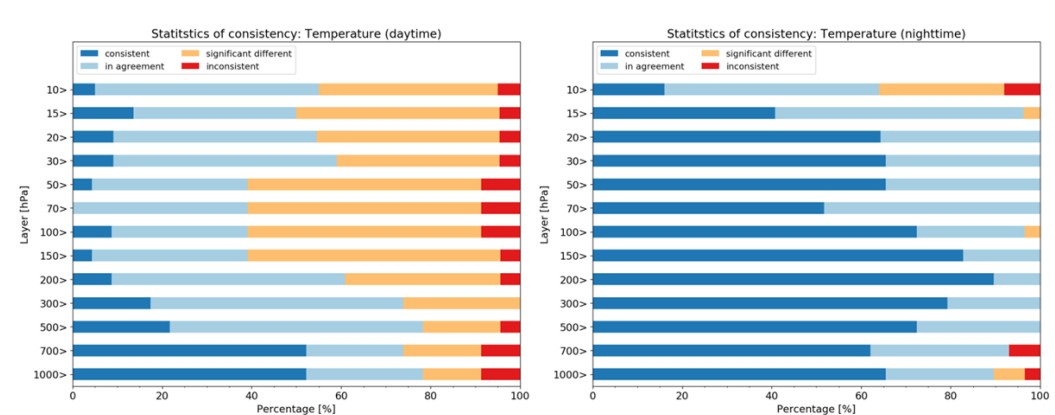

**Figure 22:** Percentages of consistency ranks "consistent," "in agreement," "significantly different", and "inconsistent" for temperature measurements between RS92 GDP and RS-11G GDP in each pressure layer for daytime (left) and nighttime (right) dual flights





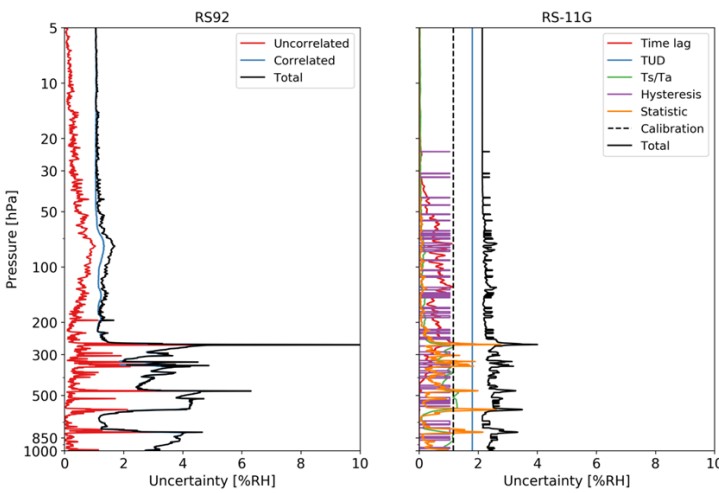

**Figure 23:** As per Fig. 21, but for RH

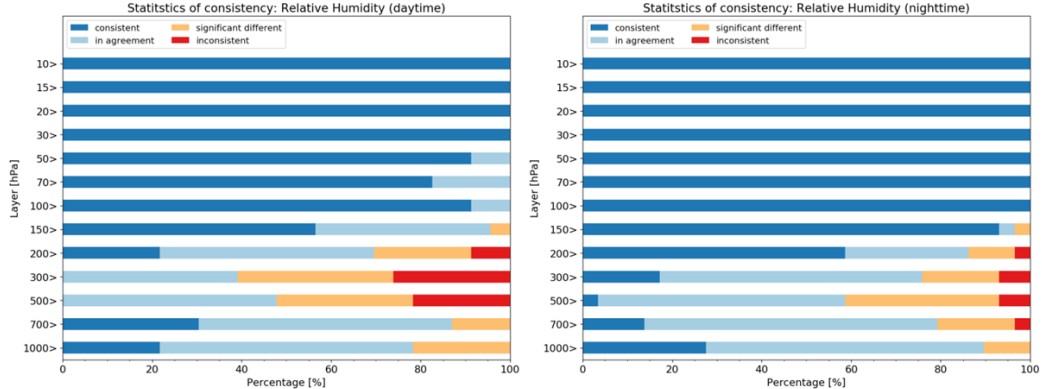

**Figure 24:** As per Fig. 22, but for RH. RH results for the stratosphere are not discussed here because measurement values for the stratosphere are considered to exceed the limit for reliable measurement.





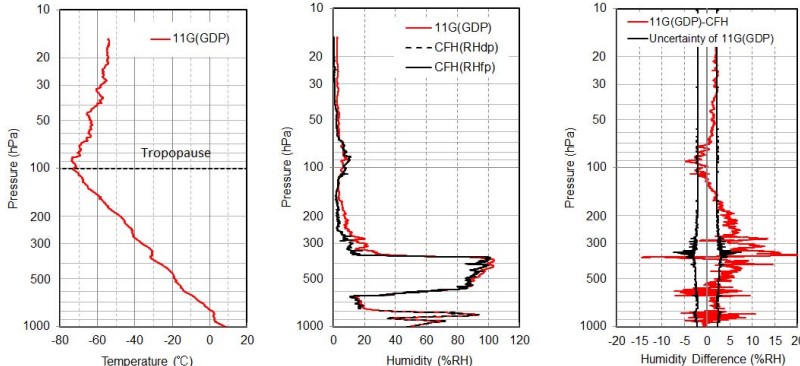

**Figure 25:** Profiles of temperature and RH/RH difference from an RS-11G and CFH comparison flight launched at 14:50 (LST) on 10 November, 2016. The panel on the left shows temperature data from RS-11G GDP. The middle panel shows the RH of RS-11G GDP (red), converted RH from CFH dewpoint temperature (black dashed line) below the height of the forced freezing point (Vömel et al., 2007), and 5 converted RH from CFH frost point temperature (black thick line) above the height of the forced freezing point. The panel on the right shows RS-11G GDP minus CFH humidity difference (red) and the overall uncertainty of RS-11G GDP (black).

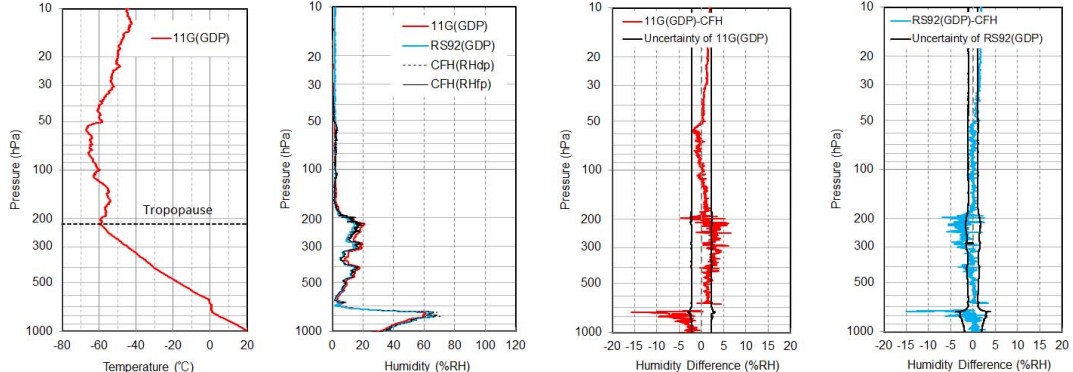

**Figure 26:** Profiles of temperature and RH/RH difference from an RS-11G, RS92, and CFH comparison flight launched at 14:50 (LST) on 10 20 April, 2018. The panel on the left shows temperature data from RS-11G GDP and the RH of RS-11G GDP (red), RS92 (light blue), converted RH from CFH dewpoint temperature (dashed black line) below the height of the forced freezing point, and converted RH from CFH frost point temperature (thick black line) above the height of the forced freezing point. The panel on the right shows RH differences for RS-11G GDP (red) and RS92 GDP (light blue) from CFH, and the overall uncertainties of RS-11G GDP and RS92 GDP (black).