# Peer review of "Comparison of the GRUAN data products for Meisei RS-11G and Vaisala RS92-SGP radiosondes at Tateno (36.06°N, 140.13°E), Japan"

_Atmospheric Measurement Techniques, 2018_

## Short Comment (SC1) · 10 Jan 2019

I wish to congratulate the authors for the clear paper and through analysis of differences between the two GRUAN products considered. In particular, the importance of measurement uncertainty produced by GRUAN approach is evident in the analysis based on Immler's formula.

Focusing on temperature, I consider two points in this discussion.

1) The paper considers the mean in each of the 13 pressure layers of Kobayashi et al. (2012) and compares the ensemble average of the two instruments for each layer. The

statistical approach is quite simple and clear. Nonetheless, state of the art literature is not considered.

*In my opinion, it should be important to deepen the literature review and cite recent papers on alternative methods for radiosonde comparisons such as the approach based on functional data analysis of Fassò et al. (2014).*

2) Figure 22 shows that the two instruments have differences with a probability much larger than stated, under zero mean Gaussian assumptions, in Tab. 4, leading to inconsistent measurements. This is especially true for daytime as noted by the authors in the abstract, in Section 5.3 and in the Summary.

Now, this may be due partly to random and correlated effects and partly to a non-Gaussian distribution of the differences. Using notation of Eq. (9), we may have that

$$d > 3u_c$$

for various (and combined) reasons:

a) The ensemble average difference, estimated in Eq.(6) and mentioned in point 1 of this discussion, is not zero. For example, the observed difference of $0.4K$ is enough to justify Tab. 4?

b) d is not Gaussian;

c) the number of samples is small.

I do not consider point c) in detail here because M is not very small and, moreover, the number of seconds per layer is relatively large.

Regarding point a), we have information given by Figure 8 where $M$ (independent) launches are averaged and each layer gives an average of possibly correlated differences ($d$). So the following question arises:

[Figure]

*Is the ensemble average difference (6) large enough to justify the inconsistency or other co-location issues are present, or measurements are not Gaussian, so the use of Immler's constants ($k = 1, 2, 3$) is not appropriate?*

I think that addressing this point using e.g. the histogram of differences ($d$) in each layer and/or other techniques to analyse the distribution of the differences ($d$) may help.

Minor points & typos:

Fig. 22 legend misspelling

Eq. (8) is a square missing?

**References**

Fassò, A, Ignaccolo, R, Madonna, F, Demoz, B. and Franco-Villoria M. (2014) Statistical modelling of collocation uncertainty in atmospheric thermodynamic profiles, Atmos. Meas. Tech., 7, 1803–1816, doi:10.5194/amt-7-1803-2014. http://www.atmos-meas-tech.net/7/1803/2014/amt-7-1803-2014.pdf

---

## Referee Comment (RC1) · Anonymous Referee #1 · 1 Feb 2019

GENERAL COMMENTS This was an interesting, straightforward, and well written paper which I enjoyed reviewing. These sorts of papers are essential to ensuring that GRUAN's operations are based on publications in the international peer-reviewed literature. This paper will be suitable for publication in AMT once my minor comments listed below have been addressed.

SPECIFIC COMMENTS

Page 1, line 13: Why were data from only 52 of the flights analysed. Why not all 87 flights? Did the other 35 not pass the quality controls for conversion to a GRUAN data product? If that is the case, perhaps that should be stated.

[Figure]

Page 1, line 15: Replace 'were 0.4 K' with 'were, on average, 0.4 K'. They were not always exactly 0.4 K lower right? A similar change needs to be made on the next two lines.

Page 1, lines 26-27: I don't know what you mean by 'Most night-time temperature differences for pressures of 10 hPa were in agreement..'. The differences were in agreement with what? Or do you mean that the measurements (not the differences) were in agreement (which I would then understand). And likewise for the next phrase about the relative humidity differences.

Page 1, lines 28-29: Similar to my comments above, when you say 'Around half of all daytime temperature differences at pressures of $\leq$ 150 hPa and relative humidity differences around the 500 hPa level were not in agreement', this is confusing. The 'differences' were not in agreement with what? I can understand the measurements not being in agreement, but a differences is one number and when you say that difference is not in agreement, it begs the question of not being in agreement with what?

Page 2, line 27: I was surprised to read that the GRUAN data product for the RS-11G was generated at Tateno. I understood that all GRUAN data processing was centralized and was done at the GRUAN Lead Centre?

Page 3, lines 24-25: You say 'and converted to create the RS-11G GRUAN data product'. Converted in what way, or converted into what? It feels like something is missing here.

Page 3, line 28: I would suggest changing 'Such errors' to 'Such external influences on the temperature measurement' since it describes more specifically what these are, and I am not sure that it is appropriate to refer to these as 'errors'. It is a correct measurement that the temperature sensor is making (there is no error), it just isn't measuring the atmospheric variable you are directly interested in i.e. the ambient air temperature.

Page 3, line 30: I don't know what a 'minima-pass filter' is. Is this the same as a low-pass filter'? If not, I think that you need to say more about what this filter is and/or cite a paper that describes the functioning of the filter. Similarly, it is not clear to me what you mean by 'minimum filter' on line 15 of page 4.

Page 4, line 17: But you say nothing about the functional form of the curve fitted to the data. Is it possible to cite an example of a correction curve to the data characterizing the temperature-dependence of the thin-film polymer RH sensors?

Page 7, lines 11-23: This information should be in a table rather.

Page 9, lines 15-16: YOu state that 'Temperature differences also influence pressure differences' but this is only true in cases where the pressure is derived from GPS altitude and temperature profiles right?

GRAMMAR AND TYPOGRAPHICAL ERRORS

This paper is very well written. Most of my suggested changes below should be interpreted very much as that i.e. suggestions.

Page 1, lines 14-15: Just a suggestion –> Replace 'The authors then quantified differences in the performance of the radiosondes using GRUAN data products' with 'Differences in the performance of the radiosondes were then quantified using the GRUAN data products'.

Page 1, lines 19-21: Again just a suggestion to write in the third person rather than referring to 'The authors' –> Replace 'The authors additionally investigated the RS-11G minus RS92-SGP difference of temperature and relative humidity based on combined uncertainties to clarify major influences behind the difference with 'Differences between the RS-11G and RS92-SGP temperature and relative humidity measurements, based on combined uncertainties, were also investigated to clarify major influences behind the differences'.

Page 1, line 22: Replace 'in RS92-SGP' with 'in the RS92-SGP'.

[Figure]

Page 1, line 23: Replace 'source for RS-11G' with 'source of uncertainty for RS-11G'.

Page 1, line 24: Replace 'and temperature-humidity' with 'and the temperature-humidity'.

Page 2, line 3: Replace 'JMA's' with 'JMA'.

Page 2, line 5: Replace 'Upper Air' with 'Upper-Air' i.e. as you have it in the abstract.

Page 2, line 10: Replace 'height' with 'altitude'.

Page 2, lines 11-12: Replace 'temperature and RH measurement has been improved' with 'the quality of the temperature and RH measurements have been improved'.

Page 2, line 14: Replace 'GRUAN will provide' with 'GRUAN is providing'.

Page 2, line 14: Delete 'for levels'.

Page 2, line 15: Replace 'will be' with 'are'.

Page 2, line 16: Replace 'will have' with 'have'.

Page 2, line 20: I would suggest changing 'habitually' to 'regularly'.

Page 2, line 25: I would suggest changing 'elevated change' to 'heightened likelihood'.

Page 3, lines 8-9: I would suggest changing 'The former is operated with a Vaisala DigiCora Sounding System III, and the latter is operated with a Meisei MGPS2' to 'The ground-station for the RS92 was a Vaisala DigiCora Sounding System, while the ground-station for the RS-11G was a Meisei MGPS2'.

Page 3, line 20: I would suggest changing 'for SHC' to 'inside the SHC' for clarity.

Page 3,line 23: Replace 'temperature measurement values' with 'temperature values'.

Page 3, line 29: Given what is displayed in Figure 1, I would suggest changing 'may be too' to 'is too'.

[Figure]

Page 3, line 31: Replace 'theoretically estimated' with 'estimated theoretically'.

Page 4, line 6: Replace 'measurement values' with 'measurements'. And likewise on line 29.

Page 4, line 12: Replace 'involves the use of' with 'uses the'.

Page 4, line 27: Replace 'height data' with 'altitude data'. And likewise on line 31.

Page 5, line 29: Replace 'is sufficient' with 'is sufficiently good'.

Page 6, lines 25-27: I would suggest changing this to 'Temporally simultaneous observations were compared, using the statistical approach adopted by Kobayashi et al. (2012), to evaluate differences in sensors and correction methods'.

Page 8, line 15: Replace 'a reference' with 'the reference'.

Page 9, line 20: When you say 'much larger than others' what does the 'others' refer to?

Page 9, line 30: Replace 'checking' with 'comparing'.

Page 10, line 13: Replace 'RH measurement' with 'the RH measurement'.

Page 10, line 33: 'few %RH degrees' sounds confusing. Is the 'degrees' necessary?

Page 11, line 4: Replace 'quantitated' with 'quantified'.

Page 12, line 9: Replace 'as per' with 'as for'.

Page 12, line 13: Replace 'the purpose of utilization' with 'use'.

Page 13, line 4: Replace 'generally corresponded' with 'generally agree well'.
* * *

---

## Referee Comment (RC2) · Anonymous Referee #2 · 3 Feb 2019

**Review of 'Comparison of the GRUAN data product for Meisei RS-11G and Vaisala RS92-SGP radiosondes at Tateno (36.06 N, 140.13 E), Japan by Eriko Kobayashi et al.**

The authors describe the comparison of a new GRUAN data product derived from the Meisei-RS11 sonde with the longstanding Vaisiala RS92-SGP product. The piece is overall well structured and written and as such publishable following relatively minor corrections.

**Major points**

My first major point would be that the fact that the two products are far more frequently found to be inconsistent with one another than would be expected by chance if both products had a comprehensive metrological traceability. We would expect only 5% of measurements to fall outside k=2 yet far more cases occur (Section 5) particularly for daytime temperatures and tropospheric humidity. While I would not expect the authors to definitively ascertain the causes of this it would be useful to: i) more explicitly flag this; ii) in the discussion more directly allude to various possible reasons; and iii) point to ways that this could in future be addressed. For the latter point the authors may wish to point to the value of repeating the analysis once a third generation RS92 product is available and whether known issues to be addressed in that product version may help or hinder the explanation of discovered differences.

My second major point would be that substantially more detail on the product derivation for the RS-11G sonde would be useful in Section 2. Is there a diagram similar to the PTU diagrams produced within GAIA-CLIM by NPL? (e.g. [http://www.gaia-clim.eu/document/product-traceability-and-uncertainty-gruan-rs92-radiosonde-temperature-product](http://www.gaia-clim.eu/document/product-traceability-and-uncertainty-gruan-rs92-radiosonde-temperature-product)). Can the traceability analysis be shown in a similar manner by expanding / modifying Figure 1? Can further details on the important sources of uncertainty and how they were derived be shown? The GAIA-CLIM approach provides a basis to do this and the tables and analysis they employ in the PTU document linked above could be used and placed in a technical appendix in the final draft, likely at little effort to support Figures 1 through 4 and provide the reader with a better expectation of the RS-11G product. Tom Gardiner of NPL may be able to help the completion of this.

Third, given the differences found between different rod configurations (Section 3) would there be value in undertaking a split analysis where the results for each rig are considered in turn? If the resulting populations are too small to justify this then I would expand Section 3 to directly state this. Otherwise it would be a reasonable question that many readers may have.

**Specific comments**:

1. In the abstract it would be useful to be clear that the departures experienced are typical values. Currently it could be read as being more exact than the results show.
2. Figures 1 through 4 the rationale for the different shaped boxes should be clearly described in the figure 1 caption and this referred to in the caption to subsequent figures.

3. To Section 2.2 it would be worth appending text that alludes to the fact that the Lead Centre plan to reprocess the RS-92 product to v3 based upon new insights. Some allusion to what the impacts of this are likely to be would, I think, be useful.
4. Page 6 paragraph starting line 5 why were there so many flights for which a GDP was not derived? Was there a pattern? Or an underlying reason why? This would be important to describe for full traceability of the analysis.
5. P. 10 line 10 and line 34 delete 'degrees'
6. In table 6 does eq. 8 refer to equation 8 in your paper (seems unlikely) or rather in Dirksen et al? If the latter please be explicit.

---

## Short Comment (SC2) · 3 Feb 2019

Regarding comments of RC2, please note that in my comment SC1, I suggest that "the histogram of differences (d) in each layer ... may help".

In fact, $P(|d| > 2*sigma) = 0.95$ is true if the differences are zero-mean Gaussian coherently with authors' lines 32-33: "Assume that $m1 = m2$ is true and that uncertainty follows normal distribution.". If the differences are zero mean, but non-Gaussian, for example, zero mean Student's t with 3 degrees of freedom, then the constant 2 is wrong and the corresponding Student's t percentile should be used.

So if, after subtracting the mean of d, the distribution of d at a certain pressure level is not approximately Gaussian, then the constant 2 is not the appropriate one.

Which would be the correct constant depends on the distribution of d. So my general consideration does not give the final answer but may help in understanding a step more on this issue.

———————————————

---

## Author Comment (AC1) · 27 Feb 2019

**Response to the comments (SC1)**

Eriko Kobayashi[1], Shunsuke Hoshino, Masami Iwabuchi, Takuji Sugidachi, Kensaku Shimizu and Masatomo Fujiwara

[1]Aerological Observatory, 1-2 Nagamine, 5 Tsukuba-shi, Ibaraki, 305-0052, Japan

eriko-kobayashi@met.kishou.go.jp

5

Thank you very much for your valuable comments and suggestions.

Our responses to your comments are as follows:

10   1)     The paper considers the mean in each of the 13 pressure layers of Kobayashi et al. (2012) and compares the ensemble average of the two instruments for each layer. The statistical approach is quite simple and clear. Nonetheless, state of the art literature is not considered.

In my opinion, it should be important to deepen the literature review and cite recent papers on alternative methods for radiosonde comparisons such as the approach based on functional data analysis of Fassò et al. (2014).

15

    -Thank you for your very useful comment. The analysis method of Fassò et al. (2014) would really be helpful for radiosonde data comparisons as an alternative approach. However, we are afraid that it will take long time for us to work on functional data analysis, and thus we would like to try it in a future work. We have added the citation of Fassò et al. (2014) as one of our future tasks.

20

   2)     Figure 22 shows that the two instruments have differences with a probability much larger than stated, under zero mean Gaussian assumptions, in Tab. 4, leading to inconsistent measurements. This is especially true for daytime as noted by the authors in the abstract, in Section 5.3 and in the Summary.

Now, this may be due partly to random and correlated effects and partly to a non-Gaussian distribution of the differences.

25   Using notation of Eq. (9), we may have that

$$d > 3u\_c$$

for various (and combined) reasons:

a) The ensemble average difference, estimated in Eq.(6) and mentioned in point 1 of this discussion, is not zero. For example, the observed difference of 0.4 K is enough to justify Tab. 4?

30   b) d is not Gaussian;

   c) the number of samples is small.

I do not consider point c) in detail here because M is not very small and, moreover, the number of seconds per layer is relatively large.

Regarding point a), we have information given by Figure 8 where M (independent) launches are averaged and each layer gives an average of possibly correlated differences (d). So the following question arises:

5   Is the ensemble average difference (6) large enough to justify the inconsistency or other co-location issues are present, or measurements are not Gaussian, so the use of Immler's constants (k=1,2,3) is not appropriate?

I think that addressing this point using e.g. the histogram of differences (d) in each layer and/or other techniques to analyse the distribution of the differences (d) may help.

10     - We followed your suggestion to create the histogram of temperature differences to confirm the distribution. Please see the following Figure R1, which shows the histogram of temperature differences for 70-50hPa and 500-300hPa pressure layers. (Figure R1 has been added to the Supplement.) Firstly, as for the possibility that sample data has outliers, we detected outliers with a criterion of d>3σ (d and σ are estimated in Eq. (3) and Eq. (7) for all data during a comparison period) and the detected data were excluded from the statistical analysis (i.e., in Fig.8 in the original manuscript).

15     Therefore, extreme outliers are not included in the results of the statistical analysis (i.e., in Fig.8 through Fig.20 in revised manuscript). However, the distribution at pressures between 500hPa and 300hPa (the bottom panels of Figure R1) shows that there are some samples with large negative differences. This is probably because there were some issues in some of the flights in the troposphere, or there may have been some cases with calibration issues. At the moment, we do not know the exact reasons.

20     The top two panels of Figure R1 are for the histogram of temperature differences for 70-50hPa pressure layer, which show that the mean of the temperature differences at daytime was about -0.4 K. The temperature differences are considered to be normally distributed. Therefore, the large temperature difference for the daytime observations may have occurred due to systematic effects. We have added this note to Section 5.3.

To investigate possible systematic effects in more detail, we need to make much more comparison observations with

25     high-performance instruments in actual flights. Also, more detailed ground checks may be necessary. These are our future tasks.

It is also possible that different rig configurations used for the comparison flights influenced the measurements, but this is not considered in the uncertainty budget in this study.

30   Minor points & typos:

Fig. 22 legend misspelling

Eq. (8) is a square missing?

-We have corrected these.

[Figure]

5    **Figure R1: Distribution of the temperature differences at pressures between 70 hPa and 50 hPa, and 500 hPa and 300 hPa for daytime.**

**Left:histogram, center:box plot, and Right:Quantile-Quantile plot.**

---

## Author Comment (AC2) · 27 Feb 2019

**Response to the comments (RC1)**

Eriko Kobayashi[1], Shunsuke Hoshino, Masami Iwabuchi, Takuji Sugidachi, Kensaku Shimizu and Masatomo Fujiwara

[1]Aerological Observatory, 1-2 Nagamine, 5 Tsukuba-shi, Ibaraki, 305-0052, Japan

eriko-kobayashi@met.kishou.go.jp

Thank you very much for your thoughtful comments and suggestions.

Our responses to your comments are as follows:

10   SPECIFIC COMMENTS

Page 1, line 13: Why were data from only 52 of the flights analysed. Why not all 87 flights? Did the other 35 not pass the quality controls for conversion to a GRUAN data product? If that is the case, perhaps that should be stated.

- The RS-11G GDP was created at Tateno; however, data for 5 flights were not used for the analysis because of the delay
15   in data processing. Therefore 82 RS-11G GDPs were available. Once an RS-11G GDP is created, quality control procedures should be taken; however, the quality control procedures have not been established and are still under consideration. On the other hand, the RS92 GDPs were created at the GRUAN Lead Centre, and quality control procedures were taken; 25 of the RS92 GDPs for the Tateno dual flights failed these procedures and were thus not available at the GRUAN data archive. The quality control procedures for the RS92 GDP are as follows (Dirksen et al.,
20   2014): The first step verifies the results of the ground check procedure; after the GRUAN corrections have been applied to raw RS92 measurements, the second step checks that profile data are within valid ranges to ensure that the estimated uncertainties of GDPs are within the manufacturer-provided uncertainties. Most of the excluded 25 RS92 GDPs have failed the second step of the quality control procedures. Among the remaining 57 sets of dual flight data, 5 were judged as outliers by the results of RS-11G RH measurements or of the temperature differences. We have revised the text in
25   section 3 as follows:

'The RS-11G GDP was created at Tateno for the analysis of this paper. However, among all the 87 dual flights involved, 5 RS-11G flight data were not used due to problems in data processing. Once an RS-11G GDP is created, quality control procedures should be taken; however, the quality control procedures have not been established and are still under consideration. Therefore, 82 RS-11G GDPs were available for this paper. Among the 82 dual flights, 25 RS92 GDPs
30   failed the quality control procedures at the GRUAN Lead Centre and were not available at the GRUAN data archive. The

quality control procedures for the RS92 GDP are as follows (Dirksen et al., 2014): The first step verifies the results of the ground check procedure; after the GRUAN corrections have been applied to raw RS92 measurements, the second step checks that profile data are within valid ranges to ensure that the estimated uncertainties of GDPs are within the manufacturer-provided uncertainties. For one of the 25 RS92 data, there was more than 1.5%RH difference between the two RH sensors at the ground check, while other 24 RS92 data did not have any problem in ground check data. Also, another one of the excluded RS92 had instrumental issues during the flight. Therefore, most of the excluded 25 RS92 have failed the second step of the quality control procedures at the GRUAN Lead Centre. It is noted that two thirds of the excluded 25 RS92 data were daytime observations, and 8 dual soundings had large differences between RS-11G and RS92 (processed at Tateno with the manufacturer's software) in temperature or RH profiles. Furthermore, 5 dual soundings were judged as outliers by the results of RS-11G RH measurements or of the temperature differences. These are the reasons why we only have 52 sets of dual flight data for the data analysis.'

Page 1, line 15: Replace 'were0.4K' with 'were, on average, 0.4K'. They were not always exactly 0.4K lower right? A similar change needs to be made on the next two lines.

-Yes, you are correct. We have revised the text as 'on average, 0.4K', and similarly revised the text for RH.

Page 1, lines 26-27: I don't know what you mean by 'Most night-time temperature differences for pressures of 10hPa were in agreement.'. The differences were in agreement with what? Or do you mean that the measurements (not the differences) were in agreement (which I would then understand). And likewise for the next phrase about the relative humidity differences.

-We have revised the text as "temperature measurements were in agreement".

Page 1, lines 28-29: Similar to my comments above, when you say 'Around half of all daytime temperature differences at pressures of $\leq 150$hPa and relative humidity differences around the 500hPa level were not in agreement', this is confusing. The 'differences' were not in agreement with what? I can understand the measurements not being in agreement, but a differences is one number and when you say that difference is not in agreement, it begs the question of not being in agreement with what?

-We meant 'measurements were not in agreement', and we have revised the text.

Page 2, line 27: I was surprised to read that the GRUAN data product for the RS-11G was generated at Tateno. I understood that all GRUAN data processing was centralized and was done at the GRUAN Lead Centre?

-Tateno is in charge of generating GRUAN data products for Meisei RS-11G. This is in part to reduce the workload of the GRUAN Lead Centre. When other GRUAN sites start to use Meisei RS-11G, Tateno is going to accept the role of generating their RS-11G GDP. We have added this explanation to Introduction.

5    Page 3, lines 24-25: You say 'and converted to create the RS-11G GRUAN data product'. Converted in what way, or converted into what? It feels like something is missing here.

-We have revised the text as 'RS-11G observation data are collected at 1-seconod intervals, and the raw data is converted into the RS-11G GRUAN data product.'

10

Page 3, line 28: I would suggest changing 'Such errors' to 'Such external influences on the temperature measurement' since it describes more specifically what these are, and I am not sure that it is appropriate to refer to these as 'errors'. It is a correct measurement that the temperature sensor is making (there is no error), it just isn't measuring the atmospheric variable you are directly interested in i.e. the ambient air temperature.

15

-We have revised this part as suggested.

Page 3, line 30: I don't know what a 'minima-pass filter' is. Is this the same as a low-pass filter'? If not, I think that you need to say more about what this filter is and/or cite a paper that describes the functioning of the filter. Similarly, it is not clear to

20    me what you mean by 'minimum filter' on line 15 of page 4.

-We have added the following explanation to the text: 'The minima-pass filtering is applied to the temperature measurements with a certain time window, which picks up only minimum values within the time window (see Kizu et al., 2018).' Also, we have revised line 15 of Page 4 (for Humidity measurements) as 'This type of wet contamination error

25    manifests as spikes in the raw RH profile; therefore, a minimum filter, which is similar to the filter for heat spikes in the temperature measurements, with a window width of pendulum frequency, is applied to the high-frequency components of raw RH data.'.

Page 4, line 17: But you say nothing about the functional form of the curve fitted to the data. Is it possible to cite an example

30    of a correction curve to the data characterizing the temperature-dependence of the thin-film polymer RH sensors?

-We have cited Figure 3.19 of Kizu et al. (2018) as follows: 'The temperature-dependence of thin-film polymer RH sensors in colder environments was evaluated under laboratory conditions by comparison with reference values from a chilled mirror hygrometer, and a correction curve was developed using the least squares method. The RH sensor has wet

biases between -60°C and 40°C, and dry biases below -60°C. Further details of the temperature dependence correction of RH sensor can be found in Kizu et al. (2018, Figure 3.19).'

Page 7, lines 11-23: This information should be in a table rather.

-We have added a new Table for information about the bins of 13 pressure layers instead of these lines.

Page 9, lines 15-16: You state that 'Temperature differences also influence pressure differences' but this is only true in cases where the pressure is derived from GPS altitude and temperature profiles right?

-This sentence indicates the pressure differences in stratosphere. We have revised the sentence as 'Temperature differences also influence pressure differences particularly in the stratosphere, because both radiosondes use temperature, relative humidity, and GPS height data to derive pressure data.'

GRAMMAR AND TYPOGRAPHICAL ERRORS

Page 1, lines 14-15: Just a suggestion –> Replace 'The authors then quantified differences in the performance of the radiosondes using GRUAN data products' with 'Differences in the performance of the radiosondes were then quantified using the GRUAN data products'.

Page 1, lines 19-21: Again just a suggestion to write in the third person rather than referring to 'The authors' –> Replace 'The authors additionally investigated the RS-11G minus RS92-SGP difference of temperature and relative humidity based on combined uncertainties to clarify major influences behind the difference with 'Differences between the RS-11G and RS92-SGP temperature and relative humidity measurements, based on combined uncertainties, were also investigated to clarify major influences behind the differences'.

Page 1, line 22: Replace 'in RS92-SGP' with 'in the RS92-SGP'.

Page 1, line 23: Replace 'source for RS-11G' with 'source of uncertainty for RS-11G'.

Page 1, line 24: Replace 'and temperature-humidity' with 'and the temperature-humidity'.

Page 2, line 3: Replace 'JMA's 'with 'JMA'.

Page2, line5: Replace 'Upper Air' with 'Upper-Air' i.e. as you have it in the abstract.

Page 2, line 10: Replace 'height' with 'altitude'.

Page 2, lines 11-12: Replace 'temperature and RH measurement has been improved' with 'the quality of the temperature and RH measurements have been improved'.

Page 2, line 14: Replace 'GRUAN will provide' with 'GRUAN is providing'.

Page 2, line 14: Delete 'for levels'.

Page 2, line 15: Replace 'will be' with 'are'.

Page 2, line 16: Replace 'will have' with 'have'.

Page 2, line 20: I would suggest changing 'habitually' to 'regularly'.

Page 2, line 25: I would suggest changing 'elevated change' to 'heightened likelihood'.

Page 3, lines 8-9: I would suggest changing 'The former is operated with a Vaisala DigiCora Sounding System III, and the latter is operated with a Meisei MGPS2' to 'The ground-station for the RS92 was a Vaisala DigiCora Sounding System, while the ground-station for the RS-11G was a Meisei MGPS2'.

Page 3, line 20: I would suggest changing 'for SHC' to 'inside the SHC' for clarity.

Page 3, line 23: Replace 'temperature measurement values' with 'temperature values'.

Page 3, line 29: Given what is displayed in Figure 1, I would suggest changing 'may be too' to 'is too'.

Page 3, line 31: Replace 'theoretically estimated' with 'estimated theoretically'.

Page 4, line 6: Replace 'measurement values' with 'measurements'. And likewise on line29.

Page 4, line 12: Replace 'involves the use of' with 'uses the'.

Page 4, line 27: Replace 'height data' with 'altitude data'. And likewise on line31.

Page 5, line 29: Replace 'is sufficient' with 'is sufficiently good'.

Page 6, lines 25-27: I would suggest changing this to 'Temporally simultaneous observations were compared, using the statistical approach adopted by Kobayashi et al.(2012), to evaluate differences in sensors and correction methods'.

Page 8, line 15: Replace 'a reference' with 'the reference'.

Page 9, line 20: When you say 'much larger than others' what does the 'others' refer to?

Page 9, line 30: Replace 'checking' with 'comparing'.

Page 10, line 13: Replace 'RH measurement' with 'the RH measurement'.

Page 10, line 33: 'few %RH degrees' sounds confusing. Is the 'degrees' necessary?

Page 11, line 4: Replace 'quantitated' with 'quantified'.

Page 12, line 9: Replace 'as per' with 'as for'.

Page 12, line 13: Replace 'the purpose of utilization' with 'use'.

Page 13, line 4: Replace 'generally corresponded' with 'generally agree well'.

-We have revised the text as suggested.

For Page 9, line 20, we have revised as 'The daytime standard deviation in spring at pressures $\leq$ 30 hPa is much larger than the standard deviation in other seasons'. Also, we have deleted the term 'degrees' for Page 10, line 33.

Thank you very much again for your valuable comments and suggestions.

**Reference**

Dirksen, R. J., Sommer, M., Immler, F. J., Hurst, D. F., Kivi, R., and Vömel, H.: Reference quality upper-air measurements: GRUAN data processing for the Vaisala RS92 radiosonde, Atmos. Meas. Tech., 7, 4463-4490, 2014.

---

## Author Comment (AC3) · 27 Feb 2019

**Response to the comments (RC2)**

Eriko Kobayashi[1], Shunsuke Hoshino, Masami Iwabuchi, Takuji Sugidachi, Kensaku Shimizu and Masatomo Fujiwara

[1]Aerological Observatory, 1-2 Nagamine, 5 Tsukuba-shi, Ibaraki, 305-0052, Japan

eriko-kobayashi@met.kishou.go.jp

Thank you very much for your thoughtful comments and suggestions.

Our responses to your comments are as follows:

10 Major points

My first major point would be that the fact that the two products are far more frequently found to be inconsistent with one another than would be expected by chance if both products had a comprehensive metrological traceability. We would expect only 5% of measurements to fall outside k=2 yet far more cases occur (Section 5) particularly for daytime temperatures and

15 tropospheric humidity. While I would not expect the authors to definitively ascertain the causes of this it would be useful to: i) more explicitly flag this; ii) in the discussion more directly allude to various possible reasons; and iii) point to ways that this could in future be addressed. For the latter point the authors may wish to point to the value of repeating the analysis once a third generation RS92 product is available and whether known issues to be addressed in that product version may help or hinder the explanation of discovered differences.

20

-Thank you very much for giving us very valuable comments. We have verified the distribution of temperature differences for daytime observation (see Figure R1) on the basis of your comments and the short comments (SC1, SC2). (Figure R1 has been added to the Supplement.) As for the temperature differences in the troposphere (bottom panels of Figure R1), although we had excluded outliers from the samples, some samples have large negative differences, which

25 are either due to some issues during the flights or to possible calibration issues. On the other hand, the histograms of the temperature differences in the stratosphere (top panels in Figure R1) are considered to be normally distributed, with a non-zero mean, and we think that extreme cases such as those in the troposphere are not included. It is speculated that the main cause of the temperature differences in the stratosphere is due to systematic effects which still have not been fully corrected by both the GDP processes. As for RH differences in the troposphere (Figure R2), we have identified

30 that the humidity sensor of RS-11G has a dry bias in the lower troposphere and a wet bias in the upper troposphere based on comparison observations with CFH. Thus, we consider that the RS-11G GDP needs to be upgraded. It is our

future task to investigate the sources of the errors by e.g., making comparison observation with high-performance instruments. Also, we agree that we need to repeat the analysis with the third generation RS92 GRUAN data product to explain more details. We have added these notes to Sections 5.3 and 5.4 as follows:

-Section 5.3 : 'Possible reasons for the fact that the percentages of "inconsistent" and "significantly different" categories are larger at pressures < 150hPa at daytime are as follows. We investigated the histogram of temperature difference and found that it is normally distributed and that the number of samples is large enough. Therefore, the temperature difference in the stratosphere at daytime is thought to be caused by unexpected systematic effects. Also, some samples showed large temperature differences (about -0.5 K) even in the troposphere, which is considered to be due either to some issues during the flights or to possible calibration problems. Further works, including comparisons with high-performance temperature instruments and additional ground checks, are required. Also, the RS92 GDP version 3 is supposed to be available in the near future (Ruud Dirksen, private communication, 2018), and it would be useful to redo the analysis with the new RS92 GDP.'

-Section 5.4 : 'There are some samples with large RH differences (more than 10 %RH), which is considered to be either due to evaporative cooling effects or related to the sensor hysteresis characteristics as mentioned in section 5.2. In addition, the authors have identified that the humidity sensor of RS-11G shows drier values in the lower troposphere and wetter values in the upper troposphere when compared to chilled-mirror hygrometer measurements as mentioned in Section 6. It is our future work to improve the RS-11G RH GDP when more intercomparison data with chilled-mirror hygrometers become available.'

[Figure]

**Figure R1: Distribution of temperature difference at pressure between 70 hPa and 50 hPa, and 500 hPa and 300 hPa for daytime.**

**Left:histogram, center:box plot, and Right:Quantile-Quantile plot.**

10

[Figure]

**Figure R2: Distribution of RH difference at pressure between 300 hPa and 200 hPa, and 1000 hPa and 700 hPa for daytime.**

**Left:histogram, center:box plot, and Right:Quantile-Quantile plot.**

My second major point would be that substantially more detail on the product derivation for the RS-11G sonde would be useful in Section 2. Is there a diagram similar to the PTU diagrams produced within GAIA-CLIM by NPL? (e.g. http://www.gaia-clim.eu/document/product-traceability-and-uncertainty-gruan-rs92-radiosonde-temperature-product).  Can the traceability analysis be shown in a similar manner by expanding / modifying Figure 1? Can further details on the important sources of uncertainty and how they were derived be shown? The GAIA-CLIM approach provides a basis to do this and the tables and analysis they employ in the PTU document linked above could be used and placed in a technical appendix in the final draft, likely at little effort to support Figures 1 through 4 and provide the reader with a better expectation of the RS-11G product. Tom Gardiner of NPL may be able to help the completion of this.

15

- We have added a diagram of traceability of the RS-11G sensors as in Figure R3 (Figure 1 in the revised manuscript). Further details of the traceability of the RS-11G sensors can be found in Section 5 of Kizu et al. (2018).

[Figure]

**Figure R3: Traceability of the temperature and RH sensors on RS-11G**

**Pink and blue ellipses indicate temperature and RH sensors, respectively. Parallelograms indicate data. The details of the correction procedures are shown in Figs. 2 through 5 in the revised manuscript.**

10

Third, given the differences found between different rod configurations (Section 3) would there be value in undertaking a split analysis where the results for each rig are considered in turn? If the resulting populations are too small to justify this then I would expand Section 3 to directly state this. Otherwise it would be a reasonable question that many readers may have.

5      -Figure R4 in this document shows temperature differences between RS-11G and RS92 for four different rig configurations. The sample size of each configuration is 11 for bamboo rod with fixed radiosondes, 21 for paper cardboard rod, 8 for bamboo rod with radiosondes hanged freely, and 17 for plastic cardboard rod. Although each sample size is not large enough, we think that it is useful to show the results for each configuration. The standard deviation for the paper cardboard rod tended to be larger than that for other rods. Also, the standard deviation with the

10     plastic cardboard rod at pressure < 30 hPa is smaller than others in daytime data. In addition, when the radiosondes are fixed to the rod directly, sensor orientation can influence the temperature measurements (Rohden et al., 2016). We are planning to write another paper on the influences of different rig configurations on the temperature measurements. We have added Figure R4 (as Figure 7 in the revised manuscript) and revised the text in Section 3 as follows:

'However, the paper cardboard rod was thicker than the bamboo rod and kept much air inside, this might have caused

15     unexpected heat flow and influenced the temperature measurements. The temperature differences investigated for each of the four different rig configurations are shown in Fig. 7. The temperature differences are averaged for each pressure layer based on the method described in Section 4.2. Note that the five outliers are not excluded in Fig. 7. The temperature difference and standard deviation for the paper cardboard rod tend to be somewhat larger than those for the bamboo and plastic rods in the lower troposphere and the lower stratosphere. In the main analysis (Figs. 8-20 in the

20     revised manuscript) three among the soundings with the paper cardboard rod were excluded because of very large temperature differences. When these three outliers are excluded, the mean difference for those with the paper cardboard rod is found to be essentially within the standard deviation for all the four configurations combined.'

[Figure]

**Figure R4: Temperature differences and standard deviation for four different rig configurations**

The temperature data were allocated to four categories, i.e., bamboo rod with fixed radiosondes (upper left two panels), paper cardboard rod (upper right two panels), bamboo rod with radiosondes hanged freely (lower left two panels), and plastic cardboard rod (lower right two panels). Red and blue lines show the results in daytime and nighttime observation, respectively. Black lines show means of temperature differences for daytime and nighttime data.

Specific comments:

1. In the abstract it would be useful to be clear that the departures experienced are typical values. Currently it could be read as being more exact than the results show.

-We have revised the text about the temperature and RH differences as follows:

The temperature measurements of RS-11G were, on average 0.4 K lower than those of RS92-SGP in the stratosphere for daytime observations. The relative humidity measurements of RS-11G were, on average 2%RH lower than those of RS92-SGP under 90–100%RH conditions, while RS-11G gave on average 5%RH higher values than RS92-SGP under ≤ 50%RH conditions.

2. Figures 1 through 4 the rationale for the different shaped boxes should be clearly described in the figure 1 caption and this referred to in the caption to subsequent figures.

-We have added the explanation for the different shaped boxes to Fig.1 through 4 (Fig. 2 through 5 in the revised manuscript).

3. To Section 2.2 it would be worth appending text that alludes to the fact that the Lead Centre plan to reprocess the RS-92 product to v3 based upon new insights.
Some allusion to what the impacts of this are likely to be would, I think, be useful.

-We have added the following sentence to Section 2.2:

While the authors used version 2 of the RS92 GDP, version 3 is supposed to be available in the near future (Ruud Dirksen, private communication, 2018), and it would be useful to redo the analysis with it.

4. Page 6 paragraph starting line 5 why were there so many flights for which a GDP was not derived? Was there a pattern? Or an underlying reason why? This would be important to describe for full traceability of the analysis.

- The quality control procedures for the RS92 GDP are as follows (Dirksen et al., 2014): The first step verifies the results of the ground check procedure; after the GRUAN corrections have been applied to raw RS92 measurements, the second step checks that profile data are within valid ranges to ensure that the estimated uncertainties of GDPs are within the manufacturer-provided uncertainties. One of the excluded soundings of RS92 had more than 1.5%RH difference

between two RH sensors at the ground check. Most of the excluded RS92 GDPs are considered to have failed the second step of the quality control procedures. Two-third of the excluded soundings were daytime observations. It is our future task to clarify the exact causes. We have added these explanations to Section 3.

5. P. 10 line 10 and line 34 delete 'degrees'

-We have revise the text as suggested.

6. In table 6 does eq. 8 refer to equation 8 in your paper (seems unlikely) or rather in Dirksen et al? If the latter please be explicit.

-It is Equation 8 in Section 5.3 or our paper. We have added the text, 'Uncertainties for frequency splitting, contamination correction, and moving averaging are associated with the use of filtering or of moving averaging, which are determined by using the standard deviation of the correction amounts.' to the caption of Table 6 (Table 7 in revised paper).
Also, we have revised the description of 'Averaging (filtering)' in Table 5 as follows: 'Derived by Eq.(8) in Section 5.3, determined by using the standard deviation of the correction amount' (Table 6 in revised paper.)

Thank you very much again for your valuable comments and suggestions.

References
Kizu N., Sugidachi T., Kobayashi E., Hoshino S., Shimizu K., Maeda R. and Fujiwara M.: Technical characteristics and GRUAN data processing for the Meisei RS-11G and iMS-100 radiosondes (GRUAN-TD-5), GRUAN Lead Centre, 2018.
Rohden, C., Sommer M. and Dirksen R.: Rigging Recommendations For Dual Radiosonde Soundings (GRUAN-TD-7). GRUAN Lead Centre, 2016.
Dirksen, R. J., Sommer, M., Immler, F. J., Hurst, D. F., Kivi, R., and Vömel, H.: Reference quality upper-air measurements: GRUAN data processing for the Vaisala RS92 radiosonde, Atmos. Meas. Tech., 7, 4463-4490, 2014.

---

## Author Comment (AC4) · 27 Feb 2019

**Response to the comments (SC2)**

Eriko Kobayashi[1], Shunsuke Hoshino, Masami Iwabuchi, Takuji Sugidachi, Kensaku Shimizu and Masatomo Fujiwara

[1]Aerological Observatory, 1-2 Nagamine, 5 Tsukuba-shi, Ibaraki, 305-0052, Japan

eriko-kobayashi@met.kishou.go.jp

5

Thank you very much for your valuable comments and suggestions.

Our responses to your comments are as follows:

10

Regarding comments of RC2, please note that in my comment SC1, I suggest that "the histogram of differences (d) in each layer ... may help".

In fact, $P(|d|>2*sigma) = 0.95$ is true if the differences are zero-mean Gaussian coherently with authors' lines 32-33: "Assume that $m1 = m2$ is true and that uncertainty follows normal distribution.". If the differences are zero mean, but non-Gaussian, for example, zero mean Student's t with 3 degrees of freedom, then the constant 2 is wrong and the corresponding Student's t percentile should be used.

So if, after subtracting the mean of d, the distribution of d at a certain pressure level is not approximately Gaussian, then the constant 2 is not the appropriate one.

Which would be the correct constant depends on the distribution of d. So my general consideration does not give the final answer but may help in understanding a step more on this issue.

- Thank you for your very useful comment. We followed your suggestion in SC1 to create the histogram of temperature differences to confirm the distribution. Please see the following Figure R1 (this figure is also attached to our reply to your comments in SC1), which shows the histogram of temperature differences for 70-50hPa pressure layer. The mean of the temperature differences at daytime was about -0.4 K in 70-50hPa pressure layer. The temperature differences are considered to be normally distributed with a non-zero mean. Thus, the large temperature difference of 70-50hPa pressure layer for the daytime observations may have occurred due to systematic effects.

30

[Figure]

**Figure R1: Distribution of the temperature differences at pressures between 70 hPa and 50 hPa for daytime.**

**Left:histogram, center:box plot, and Right:Quantile-Quantile plot.**

---

## Author Response (AR2)

**Response to the editor comments**

Eriko Kobayashi[1], Shunsuke Hoshino, Masami Iwabuchi, Takuji Sugidachi, Kensaku Shimizu and Masatomo Fujiwara

[1]Aerological Observatory, 1-2 Nagamine, 5 Tsukuba-shi, Ibaraki, 305-0052, Japan

eriko-kobayashi@met.kishou.go.jp

Thank you very much for your thoughtful comments and suggestions.

Our responses to your comments are as follows:

10 * Page 2, line 8: The Meisei iMS-100 radiosonde … --> Why are you giving this information if this type of radiosonde is not used in the manuscript? Or is the Meisei RS-11G radiosonde replaced by this type of radiosonde since September 2017.

    - The Meisei RS-11G radiosonde for routine observation at Tateno was replaced with the Meisei iMS-100 radiosonde in September 2017. We have revised the text in page 2, line 8 as follows:

    "The Meisei RS-11G radiosonde was also replaced with the Meisei iMS-100 radiosonde (Kizu et al., 2018) in September

15     2017." (Page 2, line 8 in revised paper.)

* Page 2, line 30-31: Replace with: "GRUAN sites take over some of the duties of the Lead Centre …'

    -We have revised the text as suggested.

    "GRUAN sites take over some of the duties of the Lead Centre to reduce its workload and Tateno accepted a role of

20     generating GDP of Meisei GPS sondes." (Page 2, line 30 in revised paper.)

* Page 3, line9: a Vaisala DigiCora III Sounding System, I assume

    -We have revised the text to "Vaisala DigiCora III". (Page 3, line 9 in revised paper.)

25 * Page 4, lines 3-5: The minima-pass filter is applied to the temperature measurements and only picks up minimum values within a certain time window (Kizu et al., 2018).

    -We have revised the text as suggested. (Page 4, line 3-5 in revised paper.)

* Page 4, line 11: add "are" before difficult to quantify.

30     -We have revised the text as suggested. (Page 4, line 10 in revised paper.)

* Page 5, line 13: add "(optional)" after a silicon pressure sensor because not all RS92 radiosondes have this pressure sensor implemented!

-We have revised the text as suggested. (Page 5, line 13 in revised paper.)

* Page 5, lines 22-23 vs. Page 5, lines 27-28: these two statements seem contradictory. First, you state that the raw temperature data of RS92 are corrected for heat spike errors as with RS-11G, so that the reader might conclude that you apply the same minima-pass filtering. But at the end of the paragraph, you write that heat spike errors are removed by applying a low-pass digital filter with a cut-off frequency of 0.1 Hz. Please remove this ambiguity.

-We have deleted "as with RS-11G" from page 5 lines 22-23.

"Raw temperature data are corrected for solar radiation errors and heat spike errors." (Page 5, line 22-23 in revised paper.)

* Page 6, lines 17-19: It is completely not obvious what is meant here. If the quality control procedures have not been established and are still under consideration, why do you use these 82 flight data after all? From this statement, it seems that you are using data that have not been quality controlled at all. Is this the case? Please clarify and in particular, give more details about the reason why those 5 RS-11G flight data have been removed.

* Page 6, lines 20- 30: this (new) part is poorly written. Here is a suggestion: "These quality control procedures for the RS92 GDP (Dirksen et al., 2014) consist of two steps: first, the results of the ground check procedure are verified, after applying the GRUAN corrections to the raw RS92 measurements. In a second step, it is checked that the profile data have estimated uncertainties of GDP that are within the uncertainties provided by the manufacturer. For instance, for one of the 25 rejected RS92 data, there was more than 1.5%RH difference between the RH sensors at the ground check. However, most of the excluded 25 RS92 flights failed the second step of the quality control procedures . Among these, two thirds were daytime observations. Furthermore, of the remaining 57 dual soundings, 5 dual soundings were additional blacklisted due to spurious RS-11G RH measurements or based on outlying temperature differences. So we end up with 52 sets of dual flight data for the data analysis". Please note that I dropped the part of the sentence in line 28, as it is not clear to which sample those 8 dual sounding belong (25 RS92 profiles that have been excluded? 5 dual soundings with a lot of outliers?).

-We have ordered this paragraph along your suggestions and revised as follows:

"GDPs produced from RS-11G and RS92 data between April 2015 and June 2017 were chosen for this study. Among the 87 dual flights, 25 RS92 GDPs failed the quality control procedures at the GRUAN Lead Centre and were not available at the GRUAN data archive. These quality control procedures for the RS92 GDP (Dirksen et al., 2014) consist of the following two steps. First, the results of the ground check procedure are verified. In a second step, after applying the GRUAN corrections to the raw RS92 measurements, it is checked whether the estimated uncertainties of the GDP are within the uncertainties provided by the manufacturer. For instance, for one of the 25 rejected RS92 data, there was

more than 1.5%RH difference between the two RH sensors at the ground check. However, most of the excluded 25 RS92 flights failed the second step of the quality control procedures. Among these, two thirds were daytime observations, and 8 dual soundings of the excluded 25 had large differences between RS-11G and RS92 in temperature or RH profiles (checked with processed data at Tateno with the manufacturer's software). At the time of the analysis, the RS-11G GDP was not open yet, and was created at Tateno for the analysis of this paper. In the near future, quality control procedures similar to those for RS92 GDP should be taken for the RS-11G GDP; however the quality control procedures have not been established and are still under consideration. Therefore, the quality of RS-11G was checked from temperature and RH differences from RS92 in this paper. Out of the remaining 62 dual soundings, 5 dual soundings were additional blacklisted due to spurious RS-11G RH measurements or based on outlying temperature differences. Furthermore, another 5 RS-11G GDPs were not used simply because of the delay in data preparation (not due to data quality). So we end up with 52 sets of dual flight data for the data analysis." (Page 6, line 15-30 in revised paper.)

* Page 7, lines 7-19: the described results of the analysis of the temperature differences are not so obvious from the figures to my opinion. For instance, you write "The temperature difference and standard deviation for the paper cardboard rod tend to be somewhat larger than those for the bamboo and plastic rods in the lower troposphere and the lower stratosphere." But, from the figure, could you also not conclude that paper cardboard and plastic rod having similar temperature differences? The main question from this analysis :has the rod material an impact on the temperature differences? is not convincingly answered in this analysis. Are the differences and standard deviations between the temperatures for the different rod materials significant or not? Also the last sentence of this paragraph (by the way, I think that you mean here the standard deviations of differences) is not obvious to me from the figures. You might also include to which panels of Fig 7 you are alluding to when making these statements in this paragraph.

-As you pointed out, the temperature differences for the different rig configurations were not obvious in Fig. 7. Therefore we did not separate the obtained data in the main analysis. We have revised the text in page 7, lines 7-19 as follows:

"Although the temperature differences of the four different rig configurations are similar between 500 and 50 hPa, the temperature difference for the paper cardboard rod (upper right two panels in Fig. 7) tend to be somewhat larger than those for the bamboo rod with radiosondes hanging freely (lower left two panels in Fig. 7) and plastic rod (lower right two panels in Fig. 7) in the lower troposphere (between 1000 and 500 hPa) and at pressures < 10 hPa. The source of these differences is unclear at present. In the main analysis (Figs. 8-20), three among the soundings with the paper cardboard rod were excluded because of very large temperature differences. When these three outliers are excluded, the mean difference for those with the paper cardboard rod is found to be essentially within the standard deviation of differences for all the four configurations combined. Additionally, for radiosondes with a direct rod attachment (upper left two panels and upper right two panels in Fig. 7), temperature differences can be expected due to varying sensor orientation

with respect to the position of the sun. Accordingly, the rig was replaced with a bamboo rod from which radiosondes were hung in September 2016. The latest rod, which is a plastic cardboard composite with an aluminum tape covering (Table 3) applied to reduce the effects of radiation, has been used since December 2016 based on the GRUAN recommendation (Rohden et al., 2016). The temperature differences for the different rig configurations were not significantly different and the authors did not separate data depends on the rig configurations in this study. However, an estimation of the impact of rod on observation data is important for dual soundings and it is our future tasks." (Page 7, line 9-23 in revised paper.)

* Page 9, lines 1-3: does this statement also hold for the differences, next to the standard deviations? Also, you might repeat the remaining spring daytime sample size here after removing the exceptional case.

-As you pointed out, the results for difference in spring at pressure ≤ 30 hPa is also same in this statement. We have revised the text as follows:

"The daytime difference and standard deviation in spring at pressures ≤ 30 hPa is much larger than the difference and standard deviation in other seasons, but if the exceptional case causing this, for which difference from the ensemble mean difference exceeds more than 90 m (there is 1 such samples out of 6 in total), is removed, the seasonal difference and standard deviation is very small." (Page 10, line 6-9 in revised paper.)

* Page 12, lines 14-22: another added piece of text of lower quality. First of all, you should refer to the figures that you provided in the Supplement here. Also, you do not explicitly mention the possible reasons (brought up in the short comment SC1) to which you refer in the first sentence and that you try to investigate [a] the ensemble average difference is not zero, b) the distribution is not Gaussian, c) the number of samples is small]. You should first mention those. Also replace "The authors" by "We", "is supposed to be available" with "will", and "Further works" with "Further work".

-We have revised the text as follows and add Figures of distribution (histogram and Quantile-Quantile plot) of the temperature differences between RS-11G GDP and RS92 GDP for daytime observations (Fig. 25 in revised paper).

"Possible reasons for the fact that the percentages of "inconsistent" and "significantly different" categories are larger at pressures < 150hPa at daytime are as follows: the ensemble average difference is not zero, the distribution is not Gaussian, and the number of samples is small. We investigated the histogram of temperature difference. Figure 25 shows distribution of the temperature differences between RS-11G GDP and RS92 GDP for daytime observations. We found that it is normally distributed between 70 and 50 hPa and that the number of samples is large enough. Therefore, the temperature difference in the stratosphere at daytime is thought to be caused by unexpected systematic effects. Also, some samples showed large temperature differences (about -0.5 K) even in the troposphere, which is considered to be due either to some issues during the flights or to possible calibration problems. Further work, including comparisons with high-performance temperature instruments and additional ground checks, are required. Also, the RS92 GDP

version 3 will be available in the near future (Ruud Dirksen, private communication, 2018; Sommer, 2016), and it would be useful to redo the analysis with the new RS92 GDP. " (Page 12, line 20-30 in revised paper.)

* Page 13, line 1: please specify the pressure range you are referring to for the middle troposphere.

-We have added the pressure range as follows:

"In the middle troposphere (between 500 and 200 hPa), half of RS92 GDP and RS-11G GDP values are significantly different or inconsistent." (Page 13, line 9-10 in revised paper.)

* Page 13, lines 4-7: Change to "In addition, we noted that the humidity sensor of RS-11G has a dry bias in the lower troposphere and a wet bias in the upper troposphere when compared to chilled-mirror hygrometer measurements (see Section 6). In the future, we will try to improve…"

-We have revised the text as suggested.

"In addition, we noted that the humidity sensor of RS-11G has a dry bias in the lower troposphere and a wet bias in the upper troposphere when compared to chilled-mirror hygrometer measurements (see Section 6). In the future, we will try to improve the RS-11G RH GDP when more intercomparison data with chilled-mirror hygrometers become available." (Page 13, line 12-15 in revised paper.)

* Page 13, lines 27-28. Change to "These comparisons confirm that the RS-11G GDP has a wet bias between 400 and 200 hPa, as was also common in the results shown in Fig. 11". Please repeat the origin for this effect here: "We ascribed this to …"

-We have revised the text as suggested.

"These comparisons confirm that the RS-11G GDP has a wet bias between 400 and 200 hPa, as was also common in the results shown in Fig. 11. We ascribed this to RS-11G RH sensor time-lag and sensor temperature dependence in low-temperature conditions." (Page 14, line 2-4 in revised paper.)

* Page 14, lines 5-7. Change to "We compared the GDPs using a general statistical approach based on 13 allocated pressure layers, but an analysis using the functional regression approach (e.g. Fasso et al., 2014) might be applied in the future as well.

-We have revised the text as suggested

"We compared the GDPs using a general statistical approach based on 13 allocated pressure layers, but an analysis using the functional regression approach (e.g. Fassò et al., 2014) might be applied in the future as well." (Page 14, line 13-15 in revised paper.)

* Page 14, lines 13-16. Please add the reason for this important finding.

-We have revised the text as follows:

"RS-11G GDP RH data were also evaluated based on comparison with CFH data, with results showing a wet bias in the former from CFH values between 400 and 200 hPa. The same characteristic was also observed in comparison with RS92 GDP data. The RH sensor time-lag and sensor temperature dependence in low-temperature conditions may be main factors in this wet bias." (Page 14, line 22-25 in revised paper.)

* Fig 6: please add "0.9 or" 1 m for the rod length on this figure.
-We have added "0.9 or" on Fig. 6.

* Caption of Fig 7: change to "bamboo rod with radiosondes hanging freely"
-We have revised the text as follows:

[revised manuscript text omitted]

5    **Figure 20:** As per Fig. 17, but for V

[Figure]

**Figure 21:** Temperature (a) and RH (b) profiles from the dual flight launched at 8:32 (LST) on 28th October, 2016. The panels on the left in (a) and (b) show temperature and RH profiles, respectively, from RS-11G GDP (red) and RS92 GDP (black). The middle panels show the total amount of correction. The panels on the right show differences (RS-11G GDP minus RS92 GDP) (red) and estimated uncertainties for k = 1 in blue and k = 2 in green. RH results for the stratosphere are not discussed here because measurement values for the stratosphere are considered to exceed the limit for reliable measurement.

[Figure]

**Figure 22:** As per Fig. 21 but for the flight launched at 20:30 (LST) on 4th November, 2016.

[Figure]

**Figure 23:** Uncertainty budget for temperature measurements at 00 UTC (daytime) on 28th October, 2016, for RS92 GDP (left) and RS-11G GDP (right)

[Figure]

**Figure 24:** Percentages of consistency ranks "consistent," "in agreement," "significantly different", and "inconsistent" for temperature measurements between RS92 GDP and RS-11G GDP in each pressure layer for daytime (left) and nighttime (right) dual flights

[Figure]

**Figure 25:** Distribution of the temperature differences between RS-11G GDP and RS92 GDP for daytime observations (histogram in left, box plot in center, and Quantile-Quantile plot in right.) Shown are the results for the pressure layers between 70 hPa and 50 hPa (top), between 200 hPa and 150 hPa (middle), and between 500 hPa and 300 hPa (bottom).

[Figure]

**Figure 26:** As per Fig. 23, but for RH

[Figure]

**Figure 27:** As per Fig. 24, but for RH. RH results for the stratosphere are not discussed here because measurement values for the stratosphere are considered to exceed the limit for reliable measurement.

[Figure]

**Figure 28:** Profiles of temperature and RH/RH difference from an RS-11G and CFH comparison flight launched at 14:50 (LST) on 10 November, 2016. The panel on the left shows temperature data from RS-11G GDP. The middle panel shows the RH of RS-11G GDP (red), converted RH from CFH dewpoint temperature (black dashed line) below the height of the forced freezing point (Vömel et al., 2007), and converted RH from CFH frost point temperature (black thick line) above the height of the forced freezing point. The panel on the right shows RS-11G GDP minus CFH humidity difference (red) and the overall uncertainty of RS-11G GDP (black).

[Figure]

**Figure 29:** Profiles of temperature and RH/RH difference from an RS-11G, RS92, and CFH comparison flight launched at 14:50 (LST) on 20 April, 2018. The panel on the left shows temperature data from RS-11G GDP and the RH of RS-11G GDP (red), RS92 (light blue), converted RH from CFH dewpoint temperature (dashed black line) below the height of the forced freezing point, and converted RH from CFH frost point temperature (thick black line) above the height of the forced freezing point. The panel on the right shows RH differences for RS-11G GDP (red) and RS92 GDP (light blue) from CFH, and the overall uncertainties of RS-11G GDP and RS92 GDP (black).